# Complex multi-fault rupture and triggering during the 2023 earthquake doublet in southeastern Türkiye

Chengli Liu [1] ✉, Thorne Lay [2], Rongjiang Wang[1,3], Tuncay Taymaz [4], Zujun Xie[1], Xiong Xiong[1], Tahir Serkan Irmak [5], Metin Kahraman [6] & Ceyhun Erman [4]

Two major earthquakes ($M_W$ 7.8 and $M_W$ 7.7) ruptured left-lateral strike-slip faults of the East Anatolian Fault Zone (EAFZ) on February 6, 2023, causing >59,000 fatalities and ~$119B in damage in southeastern Türkiye and northwestern Syria. Here we derived kinematic rupture models for the two events by inverting extensive seismic and geodetic observations using complex 5-6 segment fault models constrained by satellite observations and relocated aftershocks. The larger event nucleated on a splay fault, and then propagated bilaterally ~350 km along the main EAFZ strand. The rupture speed varied from 2.5-4.5 km/s, and peak slip was ~8.1 m. 9-h later, the second event ruptured ~160 km along the curved northern EAFZ strand, with early bilateral supershear rupture velocity (>4 km/s) followed by a slower rupture speed (~3 km/s). Coulomb Failure stress increase imparted by the first event indicates plausible triggering of the doublet aftershock, along with loading of neighboring faults.

The crust of Türkiye is fragmented by escape tectonics, with the Anatolian microplate displacing westward as the Arabian and African plates move northward toward the Eurasian plate (Fig. 1a). This produces active continental faulting along the right lateral strike-slip North Anatolian Fault Zone (NAFZ) and the obliquely intersecting left-lateral strike-slip East Anatolian Fault Zone (EAFZ)[1–9].

The EAFZ is a suite of primarily strike-slip faults formed by the transpressional collision between the Anatolian microplate and the Arabian Plate. The EAFZ bifurcates into a northern strand and a main strand near Celikhan (Fig. 1b). The main strand extends ~700 km with a strike averaging N60°E from the northeastern Karlıova triple junction to near the southwestern gulf of İskenderun[9–13] (Fig. 1a). Some tectonic interpretations identify a candidate Maras triple junction located near Türkoglu, joining the African, Anatolian and Arabian plates[14] with the EAFZ extending relatively straight along the Karatas-Osmaniye Fault Zone, while others extend the EAFZ southwestward along the Amanos

fault with strike N35°E[13] to a candidate Amik triple junction[15] near the northern end of the north-south striking Dead Sea Fault (DSF) (Fig. 1b). The DSF bounds the African and Arabian Plates, extending through Syria, Lebanon, Israel, and Jordan. Geodetic and geological studies indicate that the main strand of the EAFZ is divided into several distinct geometric segments by conjugate fractures, parallel faults, pull-apart basins, bends, and stepovers that may govern the size and occurrence of large earthquakes[15–17]. The northern strand of the EAFZ involves the Sürgü, Cardak, Savrun-Toprakkale, and Yumurtalik-Düzici-Iskenderun fault segments with varying strikes from E-W to N30°E[15].

The EAFZ was less seismically active than the NAFZ during the twentieth century[18,19]. However, inter-seismic geodetic coupling analysis clearly indicated that strain accumulation along the EAFZ[19–21] was capable of producing significant earthquakes, like those that struck the fault in 1513, 1795, and other events between 1822 and 1905[6,19] (Fig. 1a). Estimates of the slip rate along the main EAFZ segment range from

[1]School of Geophysics and Geomatics, China University of Geosciences, Wuhan, Hubei, China. [2]Department of Earth and Planetary Sciences, University of California Santa Cruz, Santa Cruz, CA, USA. [3]GFZ German Research Centre for Geosciences, Potsdam, Germany. [4]Department of Geophysical Engineering, The Faculty of Mines, Istanbul Technical University, Maslak 34467, Sarıyer, Istanbul, Turkey. [5]Department of Geophysical Engineering, Kocaeli University, 41380, Umuttepe, Kocaeli, Turkey. [6]Eurasian Institute of Earth Sciences, Istanbul Technical University, Maslak 34467, Sarıyer, Istanbul, Turkey. ✉e-mail: liuchengli@cug.edu.cn

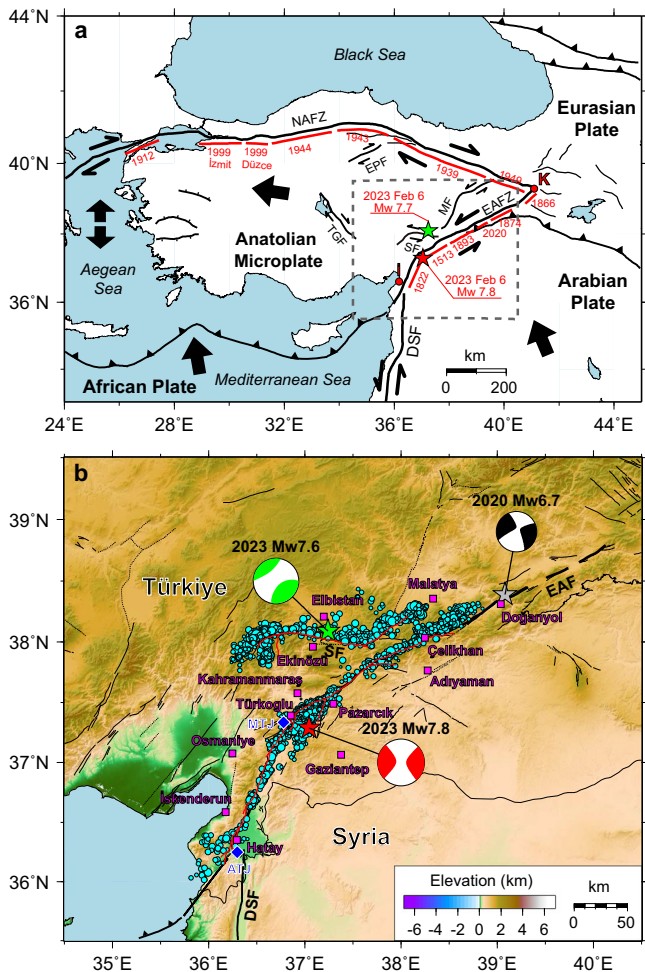

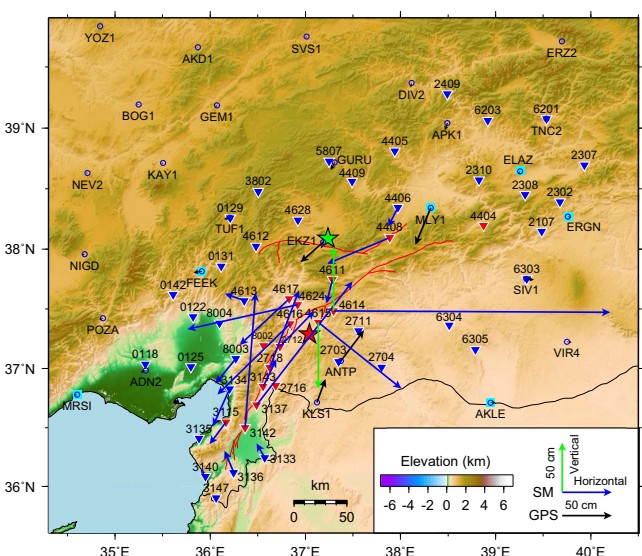

**Fig. 1 | Tectonic setting of the 2023 Türkiye earthquake doublet. a** Black thick lines show the main active faults (NAFZ north anatolian fault zone, EAFZ east anatolian fault zone, SF sürgü fault, DSF dead sea fault, EPF ezine pazari fault, TGF tuz gölü fault, MF malatya fault). Bold red lines denote the approximate rupture extent of historical events. Red K indicates the Karlıova triple junction and red İ represents İskenderun Bay. Red and green stars indicate the epicenter of the $M_W$ 7.8 and $M_W$ 7.7 earthquakes, respectively. Black thick arrows show the direction of motions between plates. The gray dashed rectangle outlines the source region of the 2023 Türkiye earthquake doublet. **b** Cyan-filled circles with a radius proportional to magnitude show the relocated aftershocks with M > 1.0. The red, green, and gray stars indicate the locations of the 2023 Türkiye earthquake doublet and the 2020 Doğanyol-Sivrice $M_W$ 6.7 event, respectively, from the AFAD-DDA catalog, and the corresponding focal mechanisms are USGS-NEIC W-phase solutions. The red lines represent fault ruptures indicated by post-earthquake satellite data[33]. Black thin lines represent active faults. The blue diamonds indicate the position of the two candidate triple junctions (MTJ maras triple junction, ATJ amik triple junction). Labeled magenta squares indicate the major cities around the source region.

**Fig. 2 | Distribution of local stations.** Inverted triangles indicate strong motion stations and cyan squares and blue circles indicate GNSS stations for the $M_W$ 7.8 event. Black vectors indicate GNSS static displacements and blue and green vectors show the horizontal and vertical coseismic displacements derived from strong-motion data, respectively. These data are used in the finite-fault inversion. The red and green stars show epicenters of the $M_W$ 7.8 and $M_W$ 7.7 events, respectively. The red lines represent positions of fault ruptures detected by post-earthquake satellite data. Inverted triangles with different colors indicate different weights used in joint inversion, with three times higher weights used for red stations.

~10 mm/y in the northeast to ~4 mm/y in the southwest where the fault system connects to the DSF[7,22–24]. On January 24, 2020, an $M_W$ 6.7 rupture struck the Doğanyol-Sivrice region of the central EAFZ main strand, northeast of the bifurcation; this was the largest event on the fault in the last 50 years prior to 2023[25–31]. The cumulative slip on the main strand of the EAFZ is modest, with estimates of 22–27 km[15,32].

On February 6, 2023, an $M_W$ 7.8 rupture (denoted the Pazarcık earthquake) initiated on a short, previously unmapped splay fault extending southward from the main strand of the EAFZ (Fig. 1b), with hypocentral parameters reported by the USGS National Earthquake Information Center (USGS-NEIC) being (37.226°N, 37.014°E, 10 km deep, at 01:17:34.332 UTC). The USGS-NEIC W-phase moment tensor

had an 81% double couple solution with a best double couple with near-vertical left lateral strike-slip with strike 228°, dip 89°, rake −1° with a seismic moment of $5.389 \times 10^{20}$ N·m ($M_{WW}$ 7.75), while the Global Centroid Moment Tensor (GCMT) solution has a best double couple with strike 51°, dip 70°, rake −4°, $M_0 = 5.8 \times 10^{20}$ N·m ($M_W$ 7.8). At 10:24:49.640 UTC, a second large event (denoted the Ekinözü earthquake) with hypocentral parameters (38.011°N, 37.196°E, 7.4 km deep), struck along the northern strand of the EAFZ with 34% double couple W-phase solution with best double couple strike 277°, dip 78°, rake 4° and seismic moment of $2.637 \times 10^{20}$ N·m ($M_{WW}$ 7.55). The GCMT solution for the Ekinözü earthquake has a best double couple with strike 264°, dip 46°, rake −9°, $M_0 = 4.53 \times 10^{20}$ N·m ($M_W$ 7.7). This pair of major earthquakes, designated as a doublet because of their similar size ($M_W$ 7.8 and $M_W$ 7.7) and close space-time proximity, produced devastating ground motions across southeastern Türkiye and northwestern Syria, responsible for >59,000 fatalities and -$119B in damage. The events ruptured complex fault networks, involving multiple fault segments resolved by satellite images[33–39]. The ground motions for both events were extensively recorded by regional strong-motion accelerometers, GNSS stations, and global broadband seismic stations (Fig. 2, Supplementary Figs. 1, 2), and the recorded signals are herein inverted for kinematic rupture models of the two major events to shed light on the ground motion generation that resulted in regional catastrophe.

## Results

### Near-fault coseismic displacements from the strong-motion data

The availability of dense near-fault strong-motion observations presents an excellent opportunity to study the detailed rupture processes of the 2023 Türkiye earthquake doublet. Near-field static deformation provides robust constraints on the slip distributions due to its insensitivity to the rupture process and precise Earth structure. To better constrain the coseismic slip distributions of the doublet events, we developed a new baseline correction method to determine static displacements from the near-fault strong-motion data, enhancing a prior

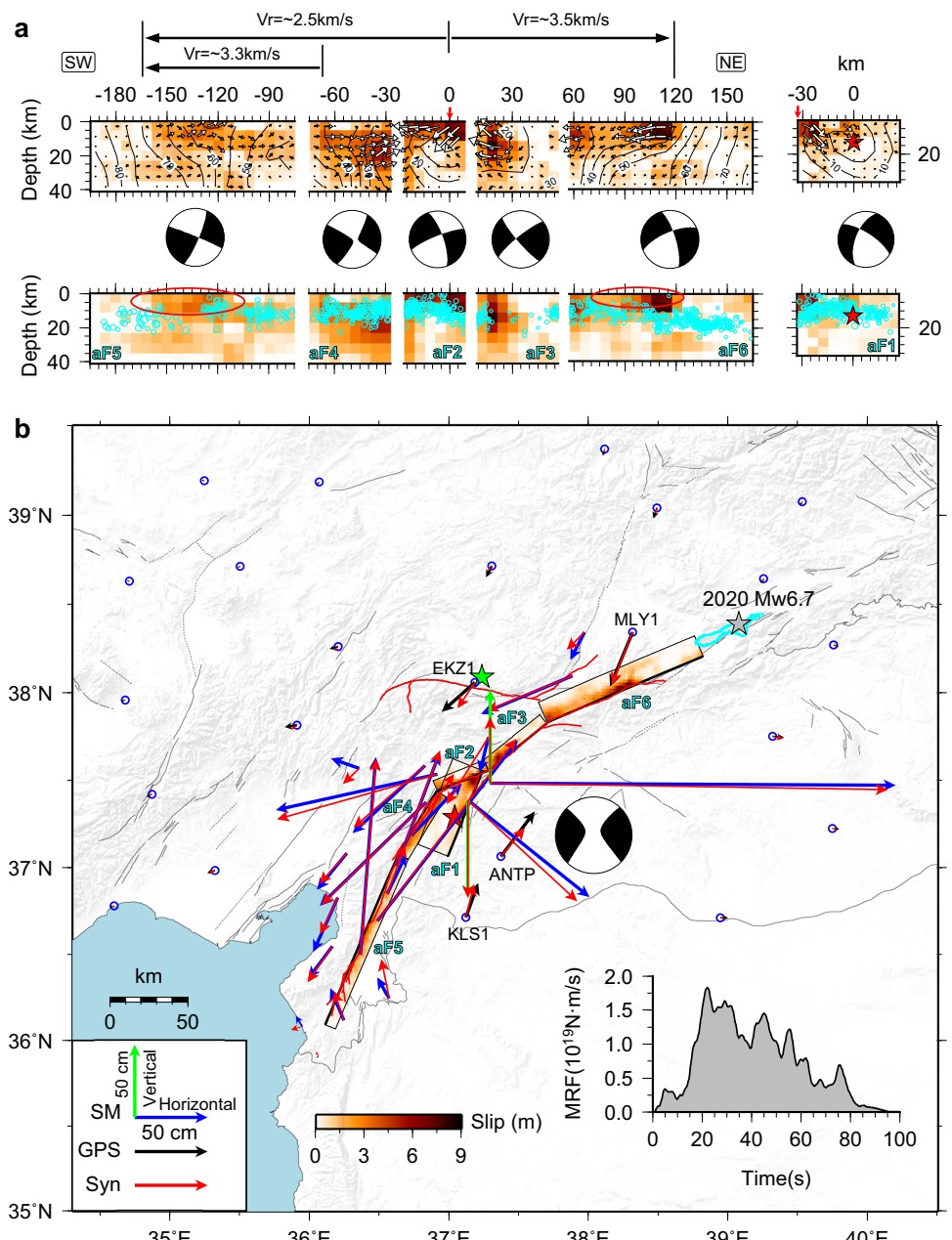

**Fig. 3 | Preferred slip model of the $M_W$ 7.8 earthquake. a** The inverted slip distribution on six fault segments; the fault parameters are listed in Supplementary Table 2. The red star locates the hypocenter on aF1. White contours indicate the slip initiation time with an interval of 5 s. White arrows indicate the direction and amplitude of the slip. Gray circles indicate aftershocks with M ≥ 2.5 less than 20 km in the fault-normal distance from each non-overlapping segment (closer near the intersecting segments), and the size is scaled by magnitude. The red ovals outline possible supershear regions, with a notable paucity of aftershocks. Focal mechanisms are the equivalent moment tensors of each fault segment. Black arrows highlight the different average rupture propagation velocities in the

northeast and southwest directions. Red vertical arrows mark the intersection of the aF1 and aF2 segments. **b** Comparison between the observed and synthetic coseismic displacements are shown in Fig. 2, and a map view of the preferred slip model. The red, green, and gray stars show epicenters of the 2023 earthquake doublet and the 2020 event, respectively. The cyan curve outlines the coseismic slip (≥0.5 m) of the 2020 Doğanyol-Sivrice $M_W$ 6.7 event[29]. The focal mechanism represents the calculated moment tensor for the composite faulting from this study. Black thin lines represent active faults. The red lines represent positions of fault ruptures detected by post-earthquake satellite data. The inset shows the moment-rate function (MRF) of the joint inversion model.

approach[40] (see Methods). The robustness and effectiveness of this new method are demonstrated by comparing estimates from strong-motion data with static coseismic GNSS displacements for several historical great earthquakes (see Code Availability). Ultimately, we successfully derived coseismic displacements at 19 near-fault strong-motion stations for the $M_W$ 7.8 event and four stations for the $M_W$ 7.7 event (Supplementary Table 1), thus compensating for a paucity of near-fault GNSS-based coseismic displacements (Fig. 2 and Supplementary Fig. 2). The recovered coseismic displacements show

prominent sinistral strike-slip characteristics, with the largest horizontal and vertical permanent values for the $M_W$ 7.8 event being 2.8 m at station 4614 and 0.6 m at station 4615, respectively (Fig. 2). These derived displacements provide valuable constraints on the coseismic slip distributions. The strong-motion derived coseismic displacements are generally consistent with horizontal displacements derived from pixel-tracking offsets of Sentinel-1 satellite radar images[37]. However, there are some differences at near-fault stations, as illustrated in Supplementary Fig. 3. Given the inherent uncertainties associated with

both approaches, such as the resolution of the pixels and the orientation error of the strong motion stations, these uncertainties inevitably contribute to differences in both magnitude and direction of the derived horizontal displacements.

## Kinematic slip model of the $M_W$ 7.8 Pazarcık earthquake

The February 6, 2023 earthquake doublet in southeastern Türkiye involves the most complex rupture evolution recorded in Türkiye throughout the last century, with backward branching[41]. The geometrical complexity of the fault ruptures was well captured by post-earthquake satellite data[33], resolving uncertainty in which fault segments ruptured and constraining the absolute location of the significant faulting with a precision of less than 1 km. Using the satellite data together with the relocated aftershock distribution[42] (Supplementary Fig. 4), we constructed a six-segment fault model (aF1–aF6) for the $M_W$ 7.8 event, with the model parameters listed in Supplementary Table 2. Based on this constrained geometry, using a well-established nonlinear finite fault inversion method (see Methods), we determined a robustly constrained space-time slip model of the $M_W$ 7.8 earthquake by joint inversion of seismological and geodetic measurements, including strong-motion data, teleseismic waveforms, static GNSS, high-rate GNSS, and the coseismic displacements derived from strong-motion observations (see data processing in the Methods section).

Our finite fault model of the $M_W$ 7.8 event indicates that the rupture began on a small fault extending southwestward from the main branch of the EAFZ, then spread onto the main branch, dipping steeply towards the northwest, with slip extending over 160 km to the northeast and terminating southwest of the rupture region of the 2020 Doğanyol-Sivrice $M_W$ 6.9 event[29] (Fig. 3). Simultaneously, rupture propagated about 180 km towards the southwestern end of the main southeast-dipping EAFZ strand, manifesting a strong bilateral rupture process. The slip distribution exhibits significant spatial heterogeneity, characterized by predominant strike-slip motion with minor occurrences of normal or thrust faulting (Fig. 3a). This pattern aligns closely with published models[36–38], highlighting the presence of lateral variations in tectonic stress, frictional properties within the crust, and intricate fault zone structures along the rupture. The estimated seismic moment $M_0 = 7.1 \times 10^{20}$ N·m ($M_W = 7.82$) during 90 s of coseismic rupture is slightly larger than the GCMT point-source solution ($M_0 = 6.1 \times 10^{20}$ N·m). The peak slip amplitude is ~8.1 m, located at the intersection of the initial fault and the main strand of the EAFZ (Fig. 3). Snapshots of the space-time slip evolution indicate that slip spread northeastward along the small branch fault during the first 10 s, and the rupture then expanded on the main strand symmetrically to the northeast and the southwest. The rupture velocities of the two propagation directions are significantly different, with the average rupture velocity in the northeast direction (~3.5 km/s) being faster than that in the southwest direction (~2.5 km/s) (Fig. 3a). This is clearly demonstrated by inversion tests with constant rupture velocities (Supplementary Figs. 5, 6). The average rupture velocity toward the northeast slightly exceeds the shear wave velocity at 5–10 km depth. Interestingly, the distribution of the relocated aftershocks shows a gap at the junction between aF3 and aF6, as well as a partial section of aF5 (Fig. 3a), indicating a large release of accumulated stress, which is consistent with typical characteristics of aftershocks distribution along supershear events[43].

The preferred slip model produces satisfactory fits for both seismic waveforms and static observations (Figs. 3b and 4, Supplementary Figs. 7, 8, and 9), suggesting a reliable model resolution. The derived coseismic displacements from near-fault strong-motion data provide helpful constraints on the slip distribution, especially in the southwestern part of the rupture. A few notable misfits are apparent in some regional waveforms, probably due to the limitations of using a 1D velocity model for calculating Green's function.

## Kinematic slip model of the $M_W$ 7.7 Ekinözü earthquake

Following the same procedure as for the $M_W$ 7.8 earthquake, we constructed a 5-fault (bF1-bF5) segment model (Supplementary Table 2) to investigate the coseismic slip model of the $M_W$ 7.7 event on the northern strand again by joint inversion of strong-motion data, teleseismic waveforms, static GNSS, high-rate GNSS and coseismic displacements derived from strong-motion observations (see Data processing in the Methods section). The preferred model shows that the rupture of the $M_W$ 7.7 event propagated bilaterally along the east-west strike direction and is primarily characterized by strike-slip offsets with significant shallow motion (Fig. 5a), and the rupture in the multiple southwest segments was more complicated than the northeastern rupture, which propagated parallel to the main strand rather than continuing on a fault extending to the main strand. The slip distribution has substantial spatial heterogeneity, with the largest slip concentrated on bF1 and bF2, presenting a complementary distribution with aftershocks (Fig. 5a). The slip on the other fault segments is relatively low, and these have dense aftershock distributions. Some available finite-fault slip models[36–38,42] show relatively smooth slip variations across much of their fault models. Despite the differences, all models are characterized by a peak slip near the epicenter while showing a minor slip along the northeastern fault segment. The maximum slip is about 11 m, located on bF1, and the total rupture duration is ~65 s. The computed seismic moment $M_0 = 5.0 \times 10^{20}$ N·m ($M_W$ 7.7), which is comparable with the GCMT solution ($4.97 \times 10^{20}$ N·m). The early (<8 s) bilateral rupture propagation had a high rupture velocity of ~4 km/s; subsequent rupture was slightly faster towards the northeast (3.0 km/s, 88% of the local shear wave velocity) than to the southwest (2.7 km/s).

The fits of static displacements and seismic waveforms of all datasets are shown in Fig. 5b and Supplementary Figs. 10, 11, and 12. The 5-fault segment model can satisfactorily explain most observations. However, despite the complexity of the fault model, there is inevitable model oversimplification and neglect of detailed 3D site effects (there is limited available information about the shallow crustal structure near the stations), and the high-frequency content of strong-motion seismic recordings are not all well explained as a result.

Simulated annealing inversions frequently exhibit slight dependence on the chosen random seeds, mainly when multiple optimal solutions exist within the model space, exhibiting indistinguishable objective function values[44]. Moreover, the varying random seeds result in distinct initial fault models and Markov chains. To address this uncertainty and explore its impact, we conducted ten inversions for each event in the earthquake doublet using different random seeds in each case. The tests indicate that large-slip distributions of the ten models for the $M_W$ 7.8 event exhibit relatively stable behavior, with consistency among the models (Supplementary Fig. 13a). In general, the standard deviation (STD) across most fault segments is negligible, with the exception of segments aF3 and aF6 (Supplementary Fig. 13b). Similarly, the STD for the $M_W$ 7.7 event is typically small compared with the average slip (Supplementary Fig. 13c), but exceptions are found in the western bF1 and bF2 fault segments (Supplementary Fig. 13d). It is suspected that the higher STD in some parts of the fault model is caused by the absence of corresponding very near-fault observations, suggesting the need for further investigation in these areas.

## Coseismic Coulomb stress changes and earthquake-triggering effects

This is a rare strike-slip major earthquake doublet with a separation interval of only 9 h; the first $M_W$ 7.8 earthquake ruptured the main branch of the EAFZ, and the second $M_W$ 7.7 earthquake ruptured the northern branch of the EAFZ. To investigate the triggering mechanism of the $M_W$ 7.7 event, we analyzed the Coulomb stress changes induced by the $M_W$ 7.8 earthquake (see Methods). Due to the significant variability in the estimated dip angle for the larger event, with faulting

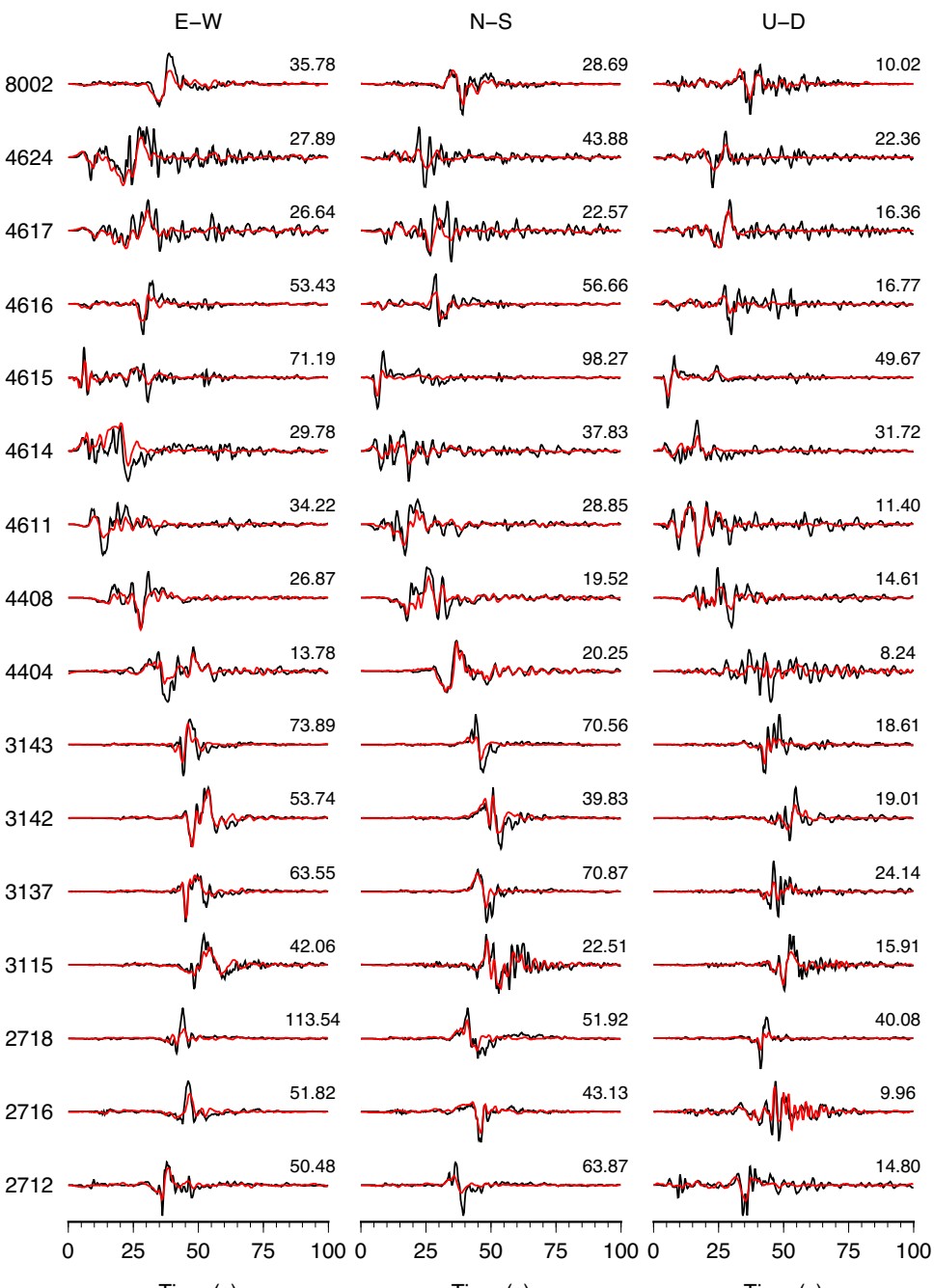

**Fig. 4 | Subsets of the strong-motion waveforms predictions.** The observed strong-motion waveforms (black lines) and synthetics (red lines) for the $M_W$ 7.8 slip model in Fig. 3. The peak ground velocities in cm/s are shown on the right, and station names are indicated on the left of each row (see Fig. 2). Comparisons with the complete set of strong-motion, high-rate GNSS, and teleseismic P and SH recordings are shown in Supplementary Figs. 6, 7, and 8.

geometries dipping to the northwest or to the southeast at angles from 42° to 86° being reported by different seismological institutes (Fig. 6), as well as the sensitivity of the results to the receiver fault parameters, we performed analyses using four different receiver fault models with varying parameters (Fig. 6). This provides a more comprehensive exploration of the triggering process by comparing the loading patterns on the initial geometry and location of the $M_W$ 7.7 event. The calculated results for 10 km depth indicate that allowing for the uncertainty in the precise geometry and slip distribution of the $M_W$ 7.8 earthquake, the Coulomb stress at the source of the $M_W$ 7.7 event increased by -0.014–0.189 MPa for four different receiver target fault geometries, in all cases exceeding the minimal earthquake triggering threshold of -0.01 MPa[45] (Fig. 6) for favorably oriented static stress

change. This is compatible with direct, albeit delayed, triggering of the back-branch rupture. It is also important to remember that triggering is complex, and larger dynamic stresses during the passage of elastic waves from the first event did not immediately trigger failure. Accumulation of pre-stress to near the failure limit and favorable orientation of the fault relative to the Coulomb stress perturbation is essential for triggering failure, with large doublet events being relatively rare as a result.

To assess the future effect on seismicity in southeastern Türkiye, we calculated the coseismic stress changes resulting from the combination of the 2023 Türkiye earthquake doublet and the 2020 Doğanyol-Sivrice $M_W$ 6.7 event[29]. We targeted three potential regions that Coulomb stress perturbations may impact: the main

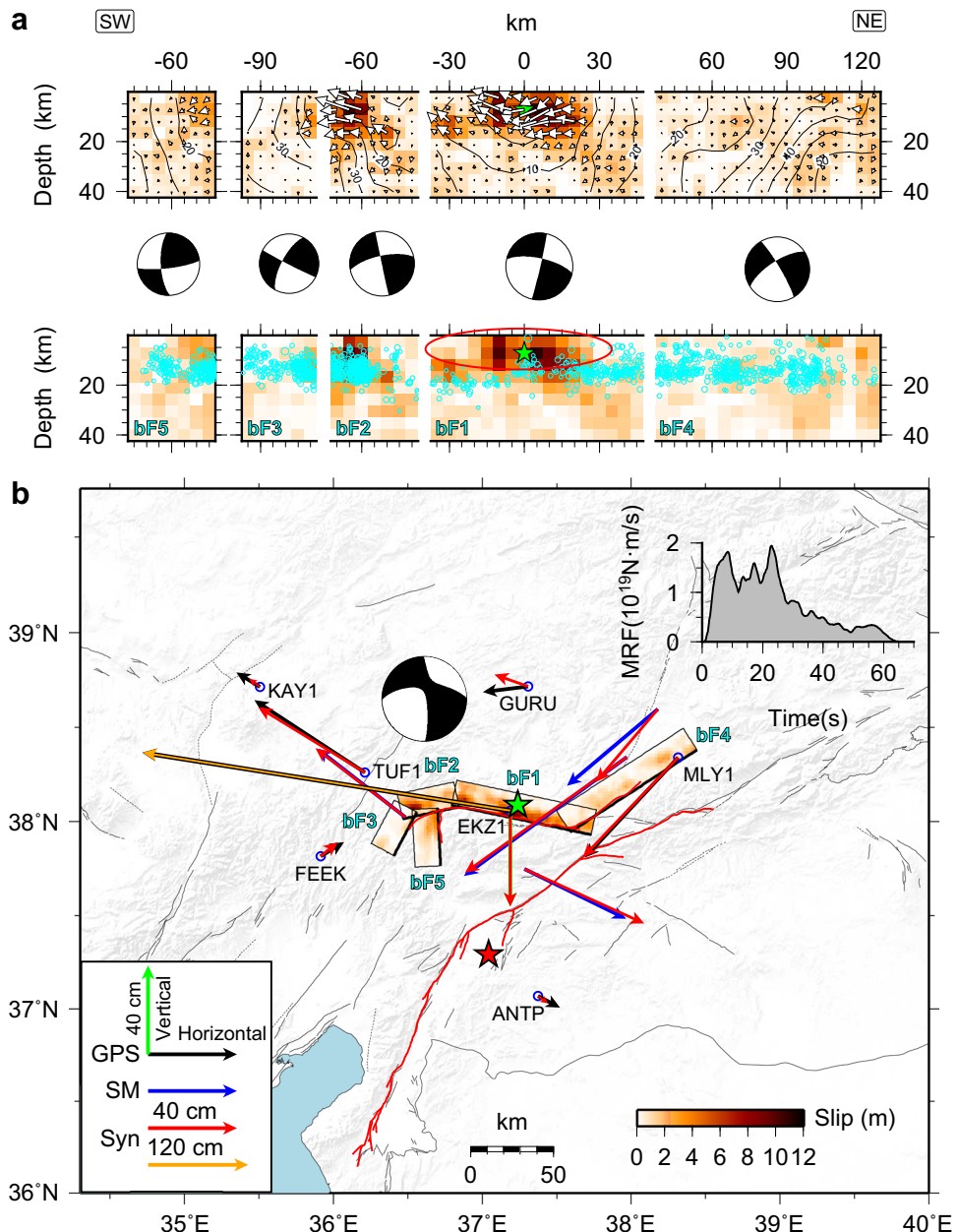

**Fig. 5 | Preferred slip model of the $M_W$ 7.7 earthquake. a, b** share the same format as Fig. 3 but for the $M_W$ 7.7 event. Waveform fits for this event are shown in Supplementary Figs. 9, 10, and 11.

northeastern strand of the EAFZ (A1), the DSF to the south (A2), and the region of the Anatolian microplate around the EAFZ (A3). The calculated Coulomb failure stress in A1 increased by up to 0.1 MPa for EAFZ receiver geometry (as shown in Fig. 7), which is concerning given that the most recent large events in the area northeast of Lake Hazar occurred in 1874 and 1866[46] (Fig. 1a). In A2, the receiver geometry of the left-lateral strike-slip Dead Sea fault, located just south of the mainshock rupture along the Amanos Fault, is calculated to have a loading increase (up to 0.1 MPa), suggesting an advance toward the next rupture. Numerous parallel strike-slip faults exist in region A3 as a result of distributed tectonic activity in the trans-pressional regime. For a receiver geometry given by the average orientation of these faults, Coulomb stress changes were computed, revealing a positive stress change zone towards the west of the northern strand. Given that significant delays ranging from years to decades between mainshocks and major aftershocks are frequently observed worldwide[45], identifying possible future rupture zones

based on the stress perturbations from the recent faulting can assist in directing mitigation efforts toward these regions.

## Discussion

In this study, we determined kinematic slip models of the 2023 Türkiye earthquake doublet in southeastern Türkiye by joint inversions of multiple seismic and geodetic datasets using faulting location constraints from satellite measurements of coseismic deformation and aftershock relocations. This reveals complex multi-fault cascading rupture processes characterized by relatively fast rupture velocities, including segments of super-shear rupture speed. Okuwaki et al.[47] found that both earthquakes in the 2023 Türkiye earthquake doublet involved supershear rupture stages, and Jia et al.[48] found on average subshear rupture for the first event and westward supershear rupture for the second. Melgar et al.[42] also estimated subshear and supershear rupture speeds for both events by inverting for slip distribution on a curved network of faults. However, our study has revealed that the

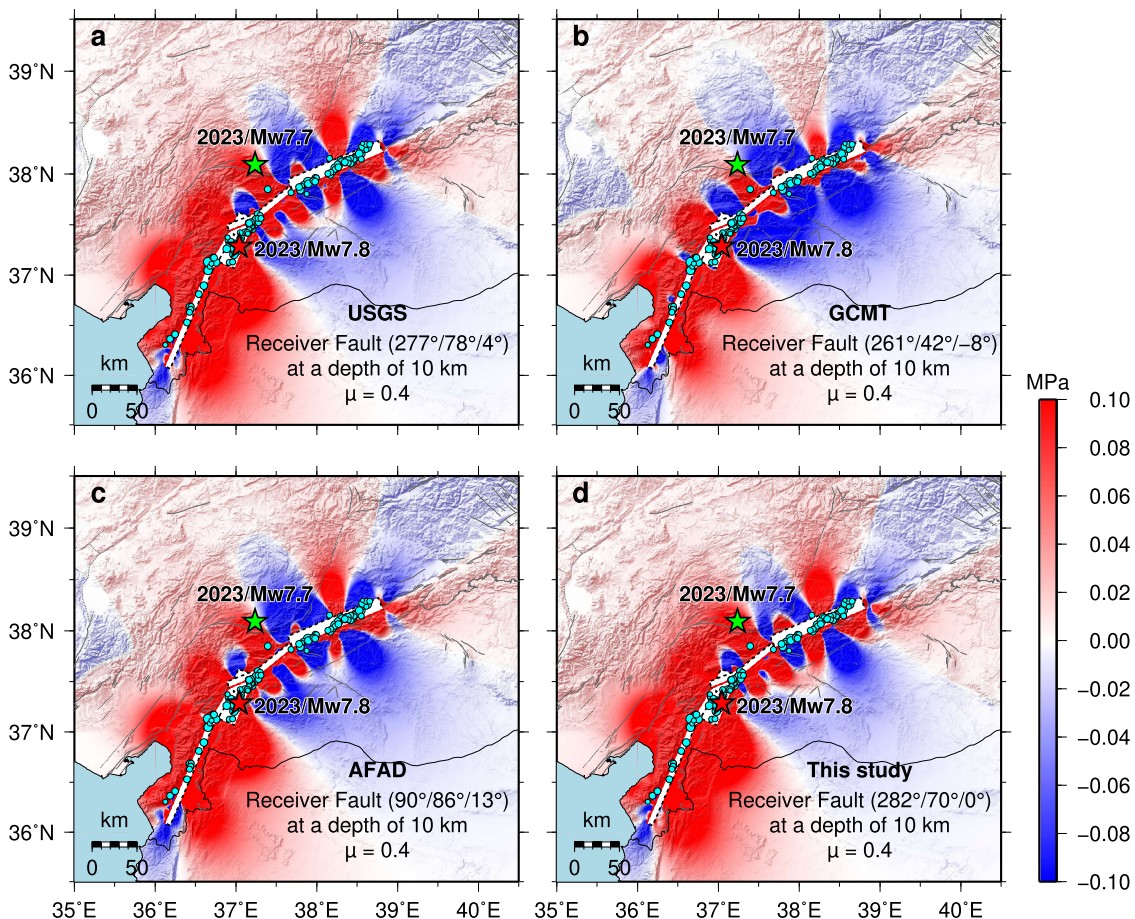

**Fig. 6 | Coulomb stress change caused by the mainshock on receiver (target) faults with the same geometry as the initial segment of the $M_W$ 7.7 event from different analyses. a, b, c**, and **d** represent the calculated results using different receiver fault parameters, at a depth of 10 km, with an effective friction coefficient of 0.4. The red and green stars show epicenters of the $M_W$ 7.8 and $M_W$ 7.7 events, respectively. The cyan-filled circles are the relocated aftershocks scaled by magnitude that occurred in the 9 h between the $M_W$ 7.8 and $M_W$ 7.7 events. Gray thin lines show the active faults.

supershear rupture of the two events occurred in somewhat distinct stages.

For the $M_W$ 7.8 event, the initial rupture velocity was relatively stable at approximately ~2.5 km/s during the first 10 s. When the rupture reached the main strand of the EAFZ and propagated northeastward, the rupture speed increased significantly within 20-40 s, reaching ~4.5 km/s between 30-40 s (Fig. 8). This high-speed rupture region was accompanied by a paucity of aftershocks (Fig. 8), and the slip patchiness suggests the possible presence of heterogeneous strengthening properties on the fault. As the rupture propagated southwestward along a relatively simple fault geometry, the rupture speed also exceeded the shear wave velocity locally, reaching ~3.8 km/s between 55-70 s but averaging about ~3.2 km/s between 40-70 s (Fig. 8).

For the $M_W$ 7.7 event, we conducted a detailed analysis of the bilateral rupture velocity of the main segment and found that the rupture velocity was relatively fast during the first 8 s, reaching ~4.0 km/s, and there is a corresponding paucity of aftershocks in the large-slip region. However, 10 s later, the rupture velocity dropped sharply on the southwest side due to a fault discontinuity in the curved western extent of the northern fault zone. In contrast, in the northeast section, the rupture velocity remained stable at around ~3.0 km/s for the time interval from 10 to 40 s, revealing non-uniformity of stress release in this fault segment (Fig. 9), which nearly parallels rather than converges with the main EAFZ strand (Fig. 1b). This suggests that high-stress buildup regions are particularly vulnerable to rapid rupture when subjected to stress perturbation. With rupture on the main strand of the EAFZ appearing to extend further northeast than on the

quasi-parallel northern strand of the EAFZ, there is no physical inconsistency with the back-branch rupture occurring where it did[41].

The observed heterogeneity in slip and rupture velocity of the two events is likely influenced by a combination of factors, including transpressional plate motion, variations in seismic coupling and fault maturity, and geometric complexities[13]. Consequently, the 2023 Türkiye earthquake doublet holds significant implications for other complex faults worldwide (e.g., San Andreas Fault in California and Kunlun Fault in north central Tibet). Given the importance of the 2023 Türkiye earthquake doublet, many studies have been and will be conducted to constrain the rupture process. Our models, which benefit from the novel inclusion of static offsets measured by numerous nearby strong motion stations, have similarities to the basic rupture distributions in prior finite-fault model determinations, but details do differ among the models. These differences arise due to different assumptions about precise model geometries (notably for dip of various fault segments), differences in data used, and differences in inversion algorithms and model parameterization.

Broadband radiated energy at teleseismic distances for the doublet events was calculated by the routine procedures[49] of the EQEnergy application of the Incorporated Research Institutions for Seismology (IRIS) (see Data Availability). For the $M_W$ 7.8 event, the total broadband teleseismic energy is estimated as $1.36 \times 10^{16}$ J, whereas for the $M_W$ 7.7 event, the broadband energy estimate is $7.29 \times 10^{15}$ J. Using the seismic moment estimates from our preferred finite-fault inversions, these give moment-scaled radiated energy estimates of $1.9 \times 10^{-5}$ and $1.5 \times 10^{-5}$, respectively. Considering the population of large

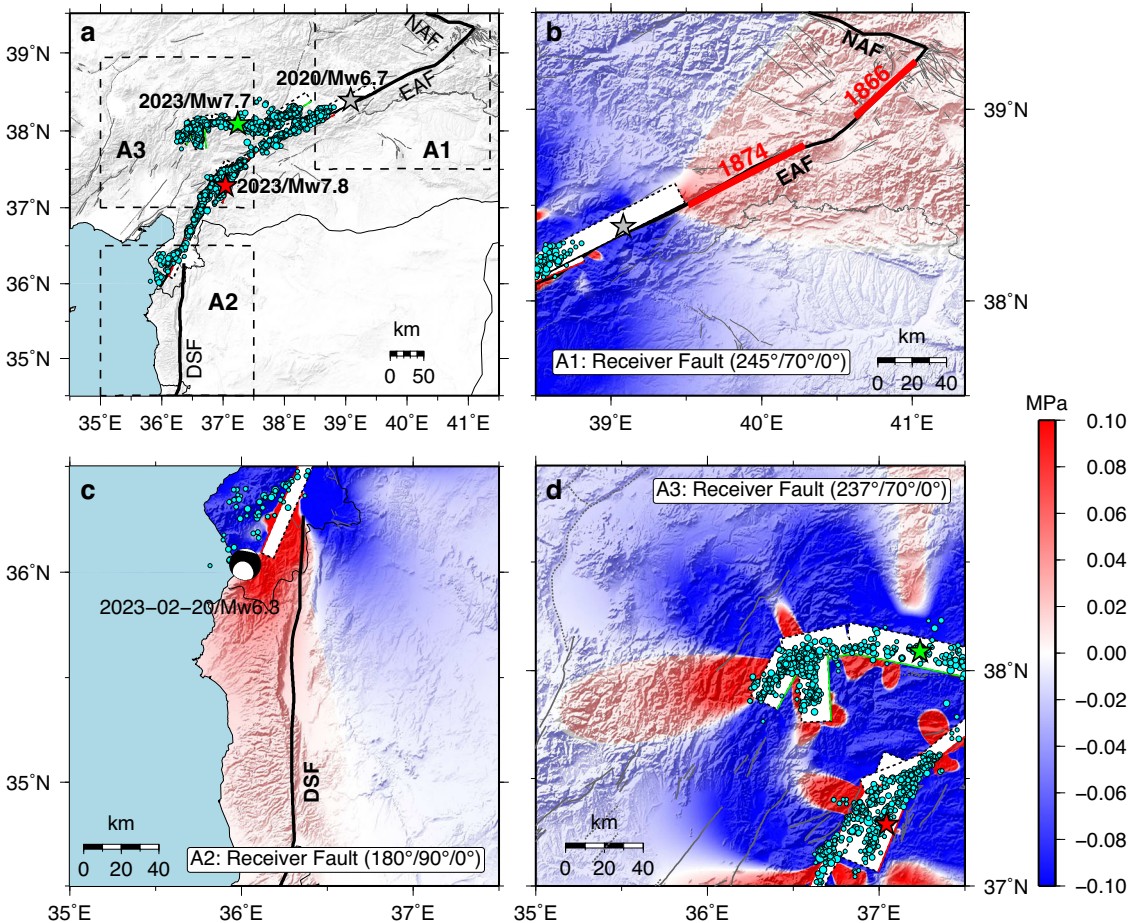

**Fig. 7 | Coulomb stress change on different receiver faults caused by the combined contributions from the 2023 earthquake doublet and the 2020 Doğanyol-Sivrice $M_W$ 6.7 event. a** The three regions considered are outlined by dashed rectangles and labeled A1, A2, and A3. Cyan-filled circles with magnitude-scaled radii show the relocated aftershocks (M > 1.0). The red and green stars show epicenters of the $M_W$ 7.8 and $M_W$ 7.7 events, respectively, and the gray star indicates the location of the 2020 Doğanyol-Sivrice $M_W$ 6.7 event. Gray thin lines show the active faults. **b, c.** The Coulomb stress changes along the northeast end of the EAF and to the south along the DSF, respectively. Bold red lines denote the approximate rupture extent of historical events. The black focal mechanism is the Febreay 20, 2023, $M_W$ 6.3 event from the USGS-NEIC W-phase solution. **d** The Coulomb stress changes in the diffuse deformation zone west of the 2023 source region. The white-filled rectangles indicate fault model segments.

strike-slip events around the world (from 1990 to 2023) with finite-fault solutions that provide seismic moments and corresponding estimates of teleseismic radiated energy (see Data Availability), establishes that these values are lower than the global mean (Fig. 10), as is the case for most events with documented supershear rupture velocity over at least portions of the rupture extent. This tendency has also been noted by Zhang et al.[50], and it may reflect relatively smoothly propagating ruptures on straight fault segments with limited slip patchiness. A rough slip distribution that frequently accelerates and decelerates the rupture enhances short-period seismic radiation, and thus roughness differences cause radiated energy differences. Unfortunately, supershear rupture and even fast sub-shear rupture produce strong directivity of lower frequency seismic radiation, enhancing shaking damage in the rupture direction. Strong sustained directivity along the southwestern Amanos fault with an average rupture velocity of 3.2 km/s during the $M_W$ 7.8 event appears to account for the massive damage in western Syria despite the slip being confined to faults within Türkiye.

The rupture on the main EAFZ during the $M_W$ 7.8 event supports the characterization of the EAFZ as connecting southwestward to the proposed Amik triple junction and the DSF, but the precise geometry of the offshore African and Anatolian plates remains ill-defined, and likely diffuse, so other secondary splay faults in the region likely have seismic potential.

The Anatolian block has previously experienced other major supershear events, such as the Izmit ($M_W$ 7.5) and Düzce ($M_W$ 7.2) earthquakes[51,52] that occurred on August 17, 1999, and November 12, 1999, respectively, resulting in severe damage and casualties. The occurrence of the 2023 Türkiye earthquake doublet has reaffirmed the findings of dynamic simulations[53], which suggested that localized supershear rupture propagation can occur near changes in fault geometry (e.g., fault bends and stepovers), even in fault segments where the initial stress field is not fully conducive to such rapid rupture.

From a tectonic perspective, the occurrence of several supershear events around the Anatolia block may be linked to the moderately high maturity and localized smooth, straight geometry of specific sections within the fault system, as well as the elevated strength and high-stress buildup resulting from the transpressional interaction of the surrounding three actively deforming plates. However, further research is necessary to fully quantify these factors to enhance our capacity to predict and mitigate the impact of such formidable events.

## Methods
### Data processing
**Teleseismic data.** We selected 40 P wave and 26 SH wave broadband waveforms for the $M_W$ 7.8 earthquake and 40 P wave and 36 SH wave for the $M_W$ 7.7 earthquake from the IRIS data management center

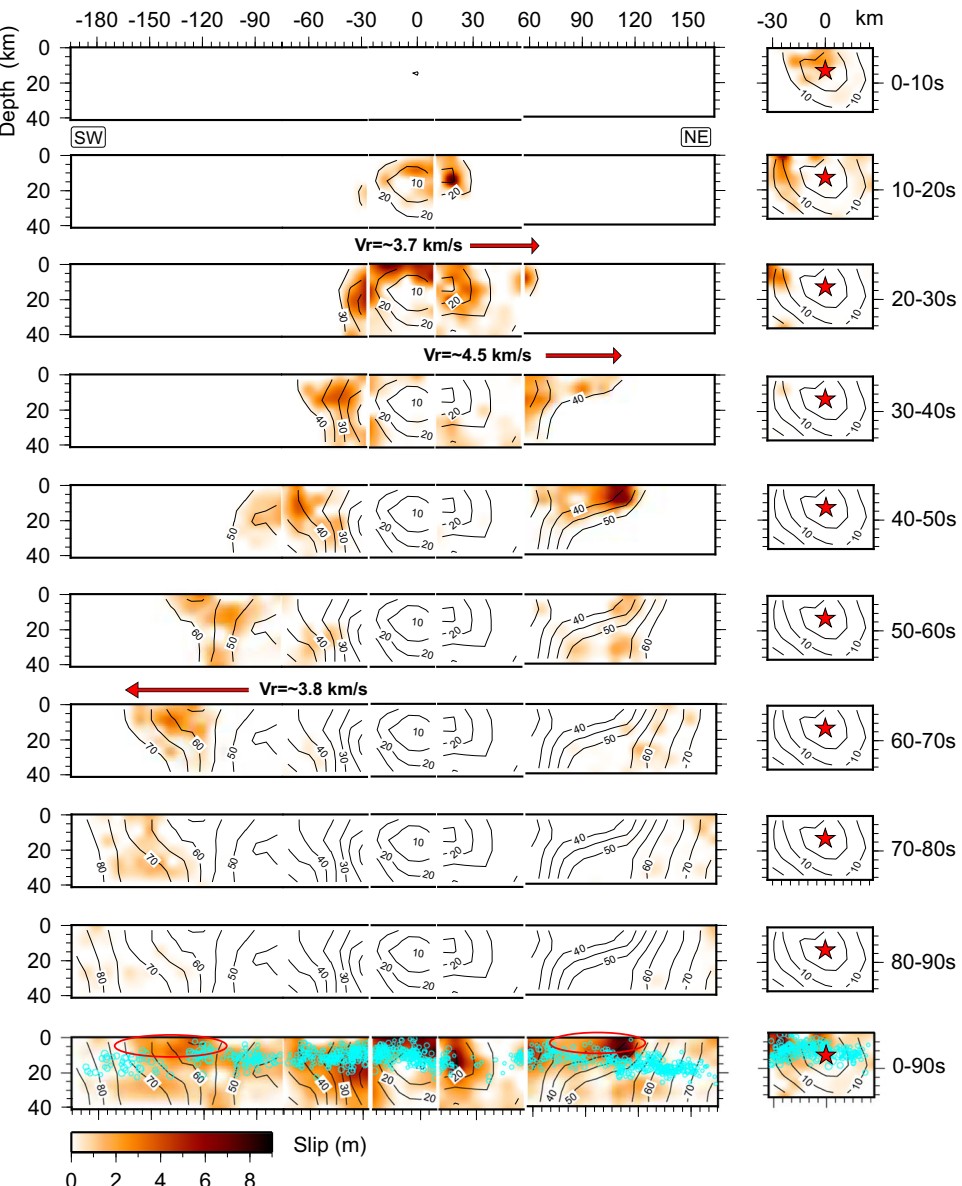

**Fig. 8 | Slip time interval snapshots that highlight the supershear rupture segments for the $M_W$ 7.8 event.** The red-filled star indicates the hypocenter. Black contours represent the slip initiation time with an interval of 5 s. Cyan circles in the lower panel indicate the relocated aftershocks, with the radius scaled by magnitude. The red ovals and red arrows highlight shallow fault stretches with inferred supershear rupture, which has a notable paucity of aftershocks.

based on high signal-to-noise ratio and well-distributed azimuthal coverage at teleseismic (30°–90°) distances (Supplementary Fig. 1). We then removed instrument responses to obtain ground displacements with durations of 100 s for the $M_W$ 7.8 event and 60 s for the $M_W$ 7.7 event, in the passband 1 s–300 s. Finally, we precisely aligned all the $P$ and $SH$ wave initial motions manually.

**Geodetic observations.** We chose the displacements time series at six GNSS stations for the $M_W$ 7.8 event and five GNSS for the $M_W$ 7.7 event, respectively, from Türkiye Ulusal Sabit GNSS Ağı (TUSAGA-Aktif) (see Data Availability), which were computed by PRIDE PPP-AR[54]. All data were re-sampled at 0.2 s intervals, and a time window of 300 s was used for the joint inversion. The first-motion arrivals of all ground displacement waveforms are hand-picked.

We also selected coseismic displacements at 29 GNSS sites (Fig. 2) for the $M_W$ 7.8 earthquake and seven GNSS sites (Supplementary Fig. 2) for the $M_W$ 7.7 earthquake from 5-min sample rate time series derived with rapid orbits by the Jet Propulsion Laboratory (see Data

Availability). Due to the relatively low precision of GNSS positions for vertical components, only the near-fault vertical component recorded at station EKZ1 was utilized in the joint inversion for the $M_W$ 7.7 earthquake.

**Strong-motion data.** Strong-motion data used in this study were recorded and provided by the Disaster and Emergency Management Presidency of Türkiye (AFAD-TK) (see Data Availability). Usually, strong-motion records include different sources of noise affecting the information to be retrieved. The most well-known problem is caused by shifts in the reference baseline, which prevent accurate ground velocity and displacement recovery through integration. The 2023 Türkiye earthquake doublet was well captured by dense strong-motion stations. To obtain high-quality velocity waveforms and coseismic permanent displacements, here, we present an updated scheme for the bi-linear baseline correction approach of Wang et al.[40]. To minimize the uncertainty of this approach, we replace the broken-line correction with a natural curve correction obtained through iterative

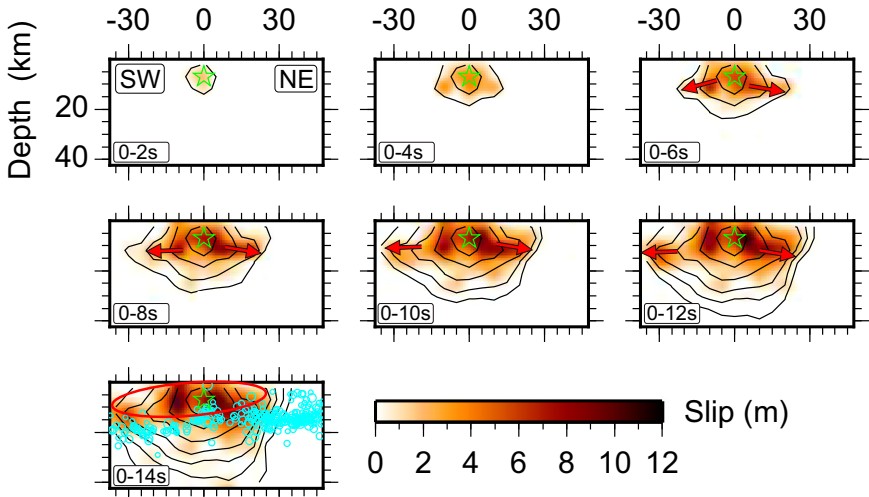

**Fig. 9 | Slip time interval snapshots that highlight the bilateral growth and early supershear rupture on segment bF1 (see Fig. 5a) for the $M_W$ 7.7 event.** The green star indicates the hypocenter. Black contours represent the rupture initiation time with an interval of 2 s. Cyan circles in the lower panel indicate the relocated aftershocks, with the radius scaled by magnitude. The red oval and red arrows indicate the region of rapid slip expansion at supershear velocity, which has a notable paucity of aftershocks.

smoothing of the uncorrected velocity seismogram to better recover the ground velocities, ground displacements, and permanent coseismic offsets. The new scheme consists of the following three steps.

Step 1. Integrate raw accelerograms to the uncorrected velocity seismograms after a pre-seismic baseline correction.

First, assume a raw accelerogram $a_{raw}(t)$ is given for time window $t \in [t_0, t_{end}]$ with the known P wave arrival $t_{pre}$ within the time window. In practice, we suggest a pre-seismic window $t_{pre} - t_0$ between 5 s and 30 s, and a generous signal window $t_{end} - t_{pre}$.

Second, estimate the pre-seismic baseline offset

$$\Delta a_{pre} = \frac{1}{t_{pre} - t_0} \int_{t_0}^{t_{pre}} a_{raw}(\tau)d\tau, \tag{1}$$

and remove it from the raw accelerogram to get an accelerogram including only seismically induced baseline errors,

$$a_0(t) = a_{raw}(t) - \Delta a_{pre}. \tag{2}$$

Third, to estimate the time when the co-seismic baseline shift is stabilized to a constant post-seismic offset, we introduce function

$$E(t) = \int_0^t |a_0(\tau)|d\tau, \tag{3}$$

and time $t_\gamma$ satisfying $E(t_\gamma) = \gamma E(t_{end})$, and assume that $t_{pst} = t_{\gamma=85\%}$ can be regarded as the time when the co-seismic baseline shift has been stabilized.

Finally, integrate $a_0(t)$ to velocity seismogram,

$$v_0(t) = \int_0^t a_0(\tau)d\tau. \tag{4}$$

Step 2. Estimate post-seismic baseline shift and the starting velocity correction.

First, calculate the post-seismic linear trend of $v_0(t)$ via least-squares regression

$$f(t) = v_{pst} + \frac{v_{end} - v_{pst}}{t_{end} - t_{pst}}(t - t_{pst}), \tag{5}$$

where $v_{pst}$ and $v_{end}$ are the start and end value of $f(t)$ at $t = t_{pst}$ and $t_{end}$, respectively.

Second, define another function

$$g(t) = \begin{cases} 0, & t_0 \le t \le t_{pre}, \\ v_0(t), & t_{pre} < t < t_{pst}, \\ w_0(t), & t_{pst} \le t \le t_{end}, \end{cases} \tag{6}$$

as the starting correction curve, where the function $w_0(t)$ is a weighted average of $v_0(t)$ and $f(t)$, i.e., the sum of right-tapered $v_0(t)$ and left-tapered $f(t)$.

Step 3. Get final velocity correction via iterative smoothing

First, smooth $g(t)$ iteratively using a small moving window, but fixing $g(t_{pre}) = 0$ and $g(t_{end}) = v_{end}$,

$$g(t) := \begin{cases} 0, & t_0 \le t \le t_{pre}, \\ \frac{1}{2\Delta t} \int_{t-\Delta t}^{t+\Delta t} g(\tau)d\tau, & t_{pre} < t < t_{pst}, \\ v_{end}, & t = t_{end}, \end{cases} \tag{7}$$

where $\Delta t$ is the time sample and $a := b$ means updating $a$ by $b$. The smoothing process is terminated when $g(t)$ has no extremum after $t_{pst}$ and at maximum only one extremum before $t_{pst}$. So $g(t)$ becomes a smooth and, in most cases, monotonic curve.

Second, set the final velocity correction curve $v_{err}(t) = 0$ for $t_0 \le t \le t_{pre}$ and $v_{err}(t) = g(t)$ for $t_{pst} \le t \le t_{end}$. For the remaining co-seismic period $t_{pre} < t < t_{pst}$, $v_{err}(t)$ needs to be estimated specially. In the case that the co-seismic and post-seismic baseline shift have the same sign, i.e., $v_{pst} \cdot (v_{end} - v_{pst}) \ge 0$, and the former is smaller than the latter, i.e., $|\frac{v_{pst}}{t_{pst} - t_{pre}}| < |\frac{v_{end} - v_{pst}}{t_{end} - t_{pst}}|$, we shift $t_{pre}$ rightward to $\max[t_{pre}, (t_{pre} + 2t_{fzc})/3]$, where $t_{fzc}$ is the time of zero-crossing of the post-seismic trend $f(t)$.

Third, construct a monotonic $v_{err}(t)$ in the co-seismic period $t_{pre} < t < t_{pst}$, which best fits $v_0(t)$ in this period in the least-squares sense.

Finally, make the baseline correction on the velocity seismogram

$$v(t) = v_0(t) - v_{err}(t) \tag{8}$$

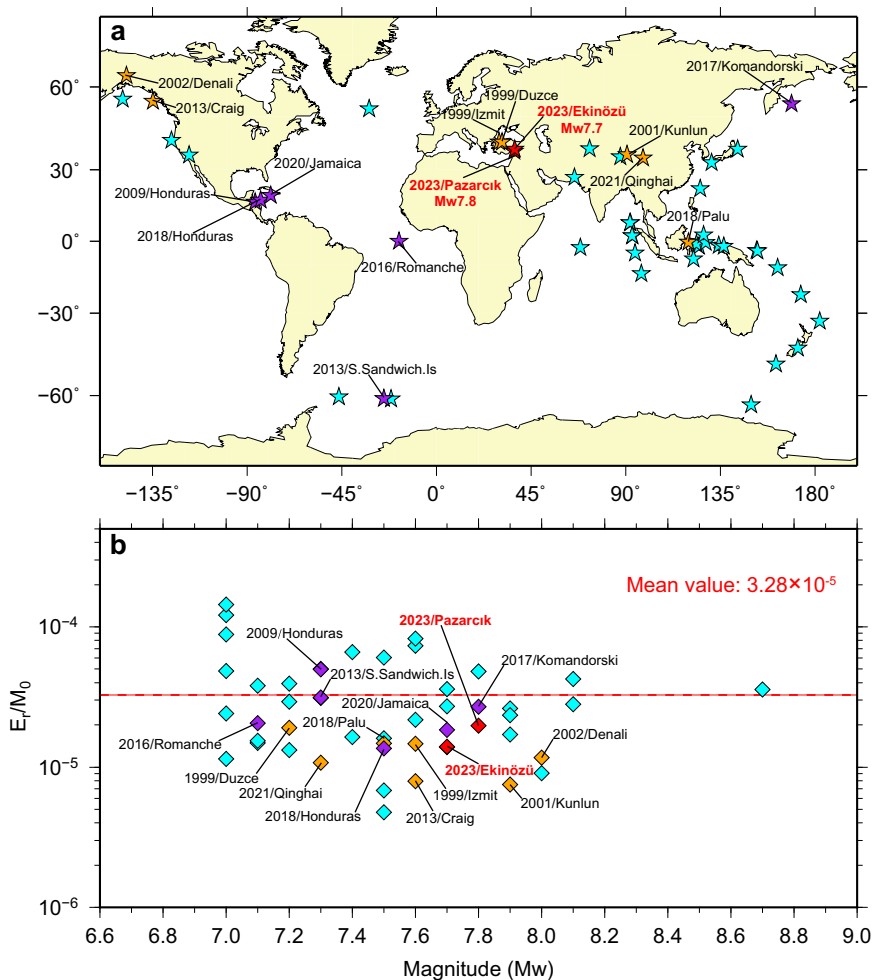

**Fig. 10 | The radiated energy/seismic moment value (E$_r$/M$_0$) for worldwide strike-slip earthquakes with M ≥ 7.0. a** The distribution of worldwide strike-slip earthquakes with M ≥ 7.0 from 1990 to 2023. The cyan stars indicate the sub-shear rupture velocity events, while the purple and orange stars indicate oceanic and continental events with portions of their ruptures modeled to be supershear, respectively. Two red stars indicate the locations of the 2023 Türkiye earthquake doublet. **b** The corresponding moment-scaled radiated energy estimates versus moment magnitude ($M_W$). The mean value ($3.28 \times 10^{-5}$) is indicated by a horizontal red dashed line.

and integrate it into the corrected displacement seismogram

$$u(t) = \int_0^t v(\tau)d\tau. \tag{9}$$

Using the correction procedure outlined above, we successfully corrected strong-motion waveforms at 52 stations for the $M_W$ 7.8 event and 26 stations for the $M_W$ 7.7 event, and obtained stable coseismic displacements at 21 near-fault stations for the $M_W$ 7.8 event and four near-fault stations for the $M_W$ 7.7 event, respectively (Supplementary Table 1). Before the joint inversion, all regional seismic waveforms were filtered with a bandpass filter of 0.02–0.5 Hz and sampled at 0.2 s intervals. The first-motion arrivals of all ground velocity waveforms are hand-picked, and a time window of 300 s was used for the joint inversion.

Verification of the ground displacement estimation from the strong-motion recordings using the foregoing procedure is provided by applications to large data sets of strong-motion and GNSS observations for the 2014 $M_W$ 8.2 Pisagua earthquake, 2014 $M_W$ 7.6 Iquique earthquake, and 2011 $M_W$ 9.0 Tohoku-Oki earthquake (see Code Availability). Favorable recovery of both horizontal and vertical displacements is achieved.

## Finite-fault inversion

A combined analysis of seismic and geodetic data is very effective in understanding the rupture process of large earthquakes. So, we utilized both data types to invert the rupture process of the $M_W$ 7.8 and $M_W$ 7.7 events using a mult-segment fault model with geometries determined by surface rupture[33] and relocated aftershocks[42] (Supplementary Fig. 4). A nonlinear finite fault inversion method is employed[55,56], which can simultaneously invert geodetic and seismic observations in the wavelet domain. The sum of L1 and L2 norms of the seismograms in different wavelets quantifies the misfit between the recorded and synthetic waveforms. Sum-squared residuals have been adopted as the evaluation criteria to measure the difference between observed and synthetic static displacements. All inversions commence with a randomly generated initial model with a total moment equal to the GCMT solution. The weight assigned to the static error is set to be equal to the waveform error, but for the statics, the weight on the coseismic displacements derived from the strong-motion data is taken as half of GNSS statics accounting for the inherent uncertainties associated with baseline correction. All inversion parameters of this earthquake doublet are presented in Supplementary Tables 2, 3. All Green's functions for both statics and waveforms are computed using a regional 1D velocity model[57].

## Coulomb failure stress

The Coulomb failure stress ($\Delta CFS$) change can be defined as[58]:$\Delta CFS = \Delta\tau + \mu'\cdot\Delta\sigma_N$, where $\Delta\tau$ and $\Delta\sigma_N$ are changes in the shear

stress and normal stress on a receiver fault, respectively, caused by the earthquake. In this study, the friction coefficient ($\mu'$) was set to 0.4 as a common choice. The values of $\Delta\tau$ and $\Delta\sigma_N$ are defined with respect to the slip and normal directions of the receiver fault, respectively. Hence, a positive value of $\Delta CFS$ indicates that the earthquake-induced stress changes push the receiver fault closer to rupture, while a negative value of $\Delta CFS$ suggests the opposite.

Using the code PSGRN/PSCMP[59], we calculated the coseismic Coulomb stress change at the location of the $M_W$ 7.7 event caused by the $M_W$ 7.8 earthquake, as well as the evolution of $\Delta CFS$ on the surrounding main faults caused by the combined stress contributions from the 2023 Türkiye earthquake doublet and the 2020 Doğanyol-Sivrice $M_W$ 6.9 event[29].

## Data availability
The facilities of IRIS Data Services, and specifically the IRIS Data Management Center, were used for access to waveforms, related metadata, and/or derived products used in this study. All teleseismic body wave records can be obtained from the Federation of Digital Seismic Networks (FDSN: https://doi.org/10.7914/SN/IU, https://doi.org/10.7914/SN/II, https://doi.org/10.7914/SN/CN, https://doi.org/10.18715/GEOSCOPE.G, https://doi.org/10.7914/SN/CU, https://doi.org/10.7914/SN/IC, https://doi.org/10.7914/SN/AV, https://doi.org/10.7914/SN/AK, https://doi.org/10.7914/SN/TA), and accessed through the IRIS data management center (http://ds.iris.edu/wilber3/find_stations/11448043). IRIS Data Services are funded through the Seismological Facilities for the Advancement of Geoscience (SAGE) Award of the National Science Foundation under Cooperative Support Agreement EAR-1851048. The strong-motion data can be obtained from https://tdvms.afad.gov.tr/continuous_data, and the raw GNSS data are from https://www.tusaga-aktif.gov.tr/. The coseismic offset measurements of GNSS for the 2023 Türkiye earthquake doublet are available from http://geodesy.unr.edu/ (http://geodesy.unr.edu/news_items/20230213/us6000jllz_final5min.txt; http://geodesy.unr.edu/news_items/20230213/us6000jlqa_final5min.txt). Broadband radiated energy at teleseismic distances is available from the EQEnergy application of the IRIS (https://ds.iris.edu/ds/products/eqenergy/). The slip models of the 2023 Türkiye earthquake doublet generated in this study can be obtained at Zenodo: https://zenodo.org/record/8232064.

## Code availability
The strong-motion baseline correction code and examples can be found at Zenodo: https://zenodo.org/record/8058010. All other calculation codes and examples used in this study are available on request from the corresponding author.

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

## Acknowledgements

The authors thank Prof. Chen Ji at the University of California, Santa Barbara, for helpful discussions. The authors thank Prof. P. Martin Mai, Dr. Jihong Liu, and Prof. Sigurjón Jónsson at King Abdullah University of Science and Technology for sharing horizontal displacements derived from pixel-tracking offsets of Sentinel-1 satellite radar images. The authors also thank Jianbao Sun and Zhaoyang Zhang at Institute of Geology, China Earthquake Administration for sharing geodetic observatons of the 2023 Türkiye earthquake doublet. C. Liu was supported by the National Science Foundation of China (No. 42222403, 42230309). T. Lay's earthquake research is supported by US National Science Foundation (Grant EAR1802364). Tuncay Taymaz acknowledges the Istanbul Technical University Research Fund (ITU-BAP) and the Alexander von Humboldt Foundation Research Fellowship Award for providing computing facilities through the Humboldt-Stiftung Follow-Up Program to support his earthquake research.

## Author contributions

C.L., Z.X., and X.X. performed the data process, finite-fault inversions, and paper writing; T.L. contributed to the model set-up, introductory material, and paper writing; R.W. processed the strong-motion data; T.T., T.S.I., M.K., and C.E. handled the HypoDD relocation of the aftershocks and contributed to tectonic material and editing.

## Competing interests

The authors declare no competing interests.
