## [Peer Review File · Nature Communications]

Complex multi-fault rupture and triggering during the 2023 earthquake doublet in southeastern TürkiyeREVIEWER COMMENTS

Reviewer #1 (Remarks to the Author):

This paper presents detailed slip models for the destructive earthquake doublet that struck southwestern Türkiye in early 2023. A few models have been published recently, but we are very much in the early stages of assessing the source of these events. A key feature of the models in this paper is that they have used an extensive set of near-field data from GPS displacements and strong motion records. These data are critical for localizing the slip in the model in space and time.

Overall, the models look good to me. The figures are mostly good, but the arrows in the panels like Figure 5a are too small to see. Overall, I think this is a valuable contribution that will advance our knowledge of the earthquake source for these events, and I think publication after minor revisions is reasonable.

I marked an annotated manuscript. There are some awkward phrasings noted, and some English corrections suggested. Awkward phrasing markings should be interpreted to mean that the author's intended meaning could not be uniquely interpreted from their words, so they need to rephrase the text to be more clear. However, overall the writing and organization are clear and the paper is easily understood.

Lines 46-51. This discussion is not so clear, and the potential triple junction points are not clearly indicated on the figures. Perhaps show the alternative geometries with different colors or line styles in Figures 1 or 2?

Lines 354-357. The use of "postpone" and "prepone" is technically correct, but "prepone" is a rarely used word. It might be simpler to use "advance" and "delay". Shifting t_{pre} to an earlier time should not cause any issues, but since t_{pre} is supposed to be the P-wave arrival time, shifting t_{pre} to a later time means that the P waves would be included in the pre-earthquake baseline window, which seems problematic. Please clarify. Also, there is a supplemental figure that shows this method applied to the Tohoku earthquake, which does not appear to be referenced in the text. The method should end with a comparison of the estimated static displacements from strong motion records to observed GPS displacements, which I think they need to use the Tohoku earthquake for. So say that in the main text, and refer to the figure.

Lines 390-398. Please add the EOS article reference provided for the UNR products, in addition to the specific URLs here.

Reviewer #2 (Remarks to the Author):

+++++

REVIEW FOR 427543

Complex multi-fault rupture and triggering 1 during the February 6, 2023, earthquake doublet in southeastern Türkiye

By Chengli Liu and co-authors

+++++

Earthquake source inversions are key to understand the physics of the earthquake rupture process, which in turn helps to unravel the underlying causes (tectonics; acting stresses; fault-zone complexities) of this particular event, but also allow to make inferences on what to possibly expect in future quakes. The earthquake sequence in Turkey of February 06, 2023, was a particular violent one in terms of shaking & damage, but also in terms of magnitudes of the two

events (M 7.8 and M 7.6 only 9 hrs later), and hence studying these earthquakes in detail is of great importance.

In this study, Liu et al use a combined data set of geodetic and seismic data for finite-fault inversions of this earthquake doublet. They also calculate the Coulomb Failure Stress (CFS) resulting from these ruptures (assuming alternative fault geometries) to understand (i) how the first earthquake may have facilitated the triggering of the 2nd event, and (ii) how the stresses have changed in the surroundings of these ruptures to potentially bring nearby faults closer to failure.

The paper is well written, the data analysis/interpretation solid, and the inverted finite-fault rupture model are in agreement with recent findings in several other studies. In this context, a few references are missing (Barbot et al, 2023, in Seismic; Mai et al, 2023, in The Seismic Record; Goldberg et al, 2023, in The Seismic Record; Petersen et al, 2023, The Seismic Record) which the authors should look at and compare their results with.

I have moderate-to-major comments, mostly editorial and for clarification, which I list below (sort of sequential).

The finite-fault inversion approach is not well documented/explained, uncertainties are not stated, the weighting of the different data sets is not explained, the fault discretization etc. This makes it impossible to assess the robustness of the inferred models.

Abstract

- + L 23 — perhaps better “>59,000 fatalities;
- + the abstract should tell which data are used and what has been done, before giving key results. This should be stated at the current end of Line 24, before rupture nucleation and propagation is presented. In some sense, Line 31-34 need to be moved forward, and then need to be expanded a bit

Introduction

- + L 82: See above comment to Line 23
- + L 92: earth → Earth
- + L 101: the authors here refer to the 2nd event to be of Mw 7.7. However, this has not been measured/inferred yet. In Line 80, they mention the USGS NEIC W-phase estimate of Mww 7.55, so this value of Mw 7.7 comes at a surprise and is confusing. Many other studies report M 7.6 for the 2nd event. Please clarify.

Kinematic Slip Models

- + L 131: “patchy heterogeneity” — somewhat awkward expression ... perhaps “spatial heterogeneity”?
- + L 133: unclear why variations in ‘minor normal and thrust faulting’ would be related to ‘lateral inhomogeneities of crustal rocks’ (assuming you mean the velocity-density structure here). Perhaps rephrase ...
- + L 154: remove ‘oversimplified’

Triggering and Seismic Hazard Evaluation

- + This section header is misleading, as no seismic hazard calculation/evaluation is done. At most, there could be a “clock advance/delay”. Please rephrase
- + L 190: add some references to the statement of ‘significant variability in dip angles’, and perhaps quantify this variability (5 degrees; 25 degrees?)
- + Line 195-201: Here it should be stated at which depth the CFS-values are calculated / reported. Is a stress increase of ~ 0.014 “significant”? And why is “ ~ 0.01 MPa” the minimal triggering threshold? This should be explained, and backup with references
- + Line 202: seismic “risk” is not assessed in this study. Not even seismic hazard. See comment above.
- + Line 212: please quantify “incremental loading increase”

Discussion

- + Line 253: I find the argument on 'significant non-uniformity of stress accumulation' somewhat confusing, in particular when it is then linked to "high stress buildup" two lines below. Non-uniformity due to small-scale fault segmentation or fault roughness? Any evidence of this due to, say, variations in moment-tensor solutions in the local background seismicity there?
- + Lines 260-280: this is interesting argument, which could be further quantified by computing the seismic-radiation efficiency. It is actually quite sobering and counterintuitive if these very damaging earthquakes had low radiation efficiency ...
- + Line 285 - 298: This seems rather speculative, in my opinion, suggesting that super-shear rupture may even be common (or the norm?) for large earthquakes on the NAF and EAF. I don't think the case of prevalence of supershear event can be made, or even should be made, based on only four documented observations.

Finite Fault Inversion

- + there is no mention of how the different datasets are weighted;
- + there is no mention of uncertainties at all;
- + there is no mention of how misfits are computed (variance reduction?)

==> this section needs to be expanded. The readers need information on robustness and uncertainties of the solutions

Coulomb Stress

- + Line 418: "we calculated the triggering" ... this is actually not correct. Only the static stress changes were computed. It is then inferred/assumed that somewhat higher stress near the hypocenter may have initiated / facilitated the 2nd event. But there is dynamic rupture simulation carried out that actually shows that ...

References

- + several recent publications that also conducted finite-fault inversions are not listed, but should be added (see above)

Figures

- + Figure 3 and 4, panels: from which fault-normal distance were aftershocks projected onto the fault plane? Up to 5 km on either side of the fault plane? More or less?
 - + Figures 8 and 9 are confusing: they show time-slices (snapshots) of the rupture process, but in each time slices, the rupture-time contours of the entire rupture is shown. Shouldn't the time-slice 0-10 sec only contain contours for the first 10 second?
- Also, I find the color-range choice suboptimal, as tiny variations in shades of blue cannot be distinguished. Regions of zero slip should be shown in white, or light gray, and thus be clearly distinct from regions with 0.5-1.0 m. This is not case here.

Supplementary Material

Figure 3 is confusing here and should be removed. Maps related to the 2011 Tohoku earthquake are not needed here

Reviewer #3 (Remarks to the Author):

Authors presented a thorough analysis on the rupture processes and possible impacts of the devastating Turkey earthquake doublet. The results suggest that both events ruptured multiple fault segments, featured supershear rupture episodes, and the first earthquake possibly aided the triggering of the second event. I like the idea of enriching the static displacement measurements

using the near-field strong motion accelerograms. However, I have some concerns with the validity of the data product and results, as well as some unclear delivery of the method approach. Acknowledging that the implication of this work is important, here I suggest a major revision, see the following points for details.

Major comments:

1. The fault geometry is one major assumption and is critical to the finite fault inversions conducted in this study. How the fault geometries, particularly the dip angles, are constructed, is unclear, which raises concerns about the validity of the findings. Additionally, the absence of satellite data and sources further compounds the issue.
2. One of the major claims is the supershear stages found during both the northeast and southwest rupture episodes of the first event. However, how to validate (quantify the uncertainty) of such estimates? To date, most finite-fault inversions and back projection-based studies suggest subshear ruptures along the EAF during the first event. These includes:

Meng et al. (<https://doi.org/10.21203/rs.3.rs-2747911/v1>)

Mai et al. (<https://doi.org/10.1785/0320230007>)

Goldberg et al. (<https://doi.org/10.1785/0320230009>)

Melgar et al. (<https://doi.org/10.26443/seismica.v2i3.387>)

Among them, Meng et al. conducted a Mach Cone analysis arguing that the rupture is predominantly subshear.

3. Authors' slip models have significant slip gaps along the northeast segment of the East Anatolian Fault Zone, and the west of the Sürgü Fault. However, this appears to contradict the satellite surface displacement data which are continuous for both events (see Mai et al. Fig. 2; Goldberger et al. Fig. 4). How certain are these slip gaps? The quickest way is to do a forward prediction of the satellite displacement field and compare with the data.

4. The conversion from strong motion accelerograms to static displacements impressively enriched the near-fault static observations. However, I am concerned with the robustness. For example, the static slip of station 4615 during the first earthquake has a 1.5m amplitude and is perpendicular to the dominant strike slip direction of the East Anatolian Fault Zone, whereas the satellite radar images do not show any significant southeast motions (Mat et al, Goldberg et al.). Authors need to make sure that such conversions are valid.

5. 0.1 MPa is still small for static triggering of the Mw 7.7 second earthquake. Is there a source of minimal earthquake triggering threshold of ~ 0.01 MPa for M7 earthquakes?

Minor comments:

1. The colorbars for slip amount are severely saturated in Figures 3, 5, 8, 9. I can't distinguish anything between 6 and 12 m.
2. Still for these figures (especially Fig. 3 and 5), the slip models (subpanel a) are inversely oriented compared with the map (subpanel b). I had a hard time to realize that the left-hand side indicates the faults to the east.

Response to reviews of "Complex multi-fault rupture and triggering during the February 6, 2023, earthquake doublet in southeastern Türkiye" by Liu et al., submitted to *Nature Communications*. The review comments are reproduced below in black type, with our responses and indications of how we have revised the manuscript to address the reviews indicated in blue.

REVIEWER COMMENTS

Reviewer #1 (Remarks to the Author):

This paper presents detailed slip models for the destructive earthquake doublet that struck southwestern Türkiye in early 2023. A few models have been published recently, but we are very much in the early stages of assessing the source of these events. A key feature of the models in this paper is that they have used an extensive set of near-field data from GPS displacements and strong motion records. These data are critical for localizing the slip in the model in space and time.

Overall, the models look good to me. The figures are mostly good, but the arrows in the panels like Figure 5a are too small to see. Overall, I think this is a valuable contribution that will advance our knowledge of the earthquake source for these events, and I think publication after minor revisions is reasonable.

We modified Figure 5a to make the arrows clearer.

I marked an annotated manuscript. There are some awkward phrasings noted, and some English corrections suggested. Awkward phrasing markings should be interpreted to mean that the author's intended meaning could not be uniquely interpreted from their words, so they need to rephrase the text to be more clear. However, overall the writing and organization are clear and the paper is easily understood.

We appreciate the reviewer's comments and suggestions, and we reworded the awkward phrasings in the revised manuscript. All comments in the annotated manuscript were addressed.

Lines 46-51. This discussion is not so clear, and the potential triple junction points are not clearly indicated on the figures. Perhaps show the alternative geometries with different colors or line styles in Figures 1 or 2?

We clarified the discussion and revised Figure 1b by adding two diamond symbols to locate the potential triple junctions.

Lines 354-357. The use of "postpone" and "prepone" is technically correct, but "prepone" is a rarely used word. It might be simpler to use "advance" and "delay". Shifting t_{pre} to an earlier time should not cause any issues, but since t_{pre} is supposed to be the P-wave arrival time, shifting t_{pre} to a later time means that the

P waves would be included in the pre-earthquake baseline window, which seems problematic. Please clarify. Also, there is a supplemental figure that shows this method applied to the Tohoku earthquake, which does not appear to be referenced in the text. The method should end with a comparison of the estimated static displacements from strong motion records to observed GPS displacements, which I think they need to use the Tohoku earthquake for. So say that in the main text, and refer to the Figure.

Thanks for the suggestion. A significant baseline shift does not necessarily start with the P-wave arrival. To possibly avoid over-corrections within the coseismic period, it is sometimes necessary to delay t_{pre} , particularly when the average of transient baseline shift appears with the same sign as but smaller in magnitude than the permanent one. Of course, the way to delay t_{pre} that we suggested is empirically-based. We revised the descriptions of the method for strong-motion baseline correction to make it more straightforward. We removed the supplemental Fig. 3 showing an application to Tohoku event data to avoid confusion, but added a new database in the section of Code Availability (<https://zenodo.org/record/8058010>) to demonstrate the effectiveness of the new correction method.

Lines 390-398. Please add the EOS article reference provided for the UNR products, in addition to the specific URLs here.

We added it as suggested.

We thank the reviewer for their comments, and the manuscript is improved by our revisions to address them. This is noted in the revised acknowledgments.

Reviewer #2 (Remarks to the Author):

+++++

REVIEW FOR 427543

Complex multi-fault rupture and triggering 1 during the February 6, 2023, earthquake doublet in southeastern Türkiye

By Chengli Liu and co-authors

+++++

Earthquake source inversions are key to understand the physics of the earthquake rupture process, which in turn helps to unravel the underlying causes (tectonics; acting stresses; fault-zone complexities) of this particular event, but also allow to make inferences on what to possibly expect in future quakes. The earthquake sequence in Turkey of February 06, 2023, was a particular violent one in terms of

shaking & damage, but also in terms of magnitudes of the two events (M 7.8 and M 7.6 only 9 hrs later), and hence studying these earthquakes in detail is of great importance.

In this study, Liu et al use a combined data set of geodetic and seismic data for finite-fault inversions of this earthquake doublet. They also calculate the Coulomb Failure Stress (CFS) resulting from these ruptures (assuming alternative fault geometries) to understand (i) how the first earthquake may have facilitated the triggering of the 2nd event, and (ii) how the stresses have changed in the surroundings of these ruptures to potentially bring nearby faults closer to failure.

The paper is well written, the data analysis/interpretation solid, and the inverted finite-fault rupture model are in agreement with recent findings in several other studies. In this context, a few references are missing (Barbot et al, 2023, in Seismic; Mai et al, 2023, in The Seismic Record; Goldberg et al, 2023, in The Seismic Record; Petersen et al, 2023, The Seismic Record) which the authors should look at and compare their results with.

We added the recently published references and made the comparisons (see below) as suggested in the revised manuscript.

“The slip distribution exhibits significant spatial heterogeneity, characterized by predominant strike-slip motion with minor occurrences of normal or thrust faulting (Fig. 3a). This pattern aligns closely with published models³⁶⁻³⁸, highlighting the presence of lateral variations in tectonic stress, frictional properties within the crust, and intricate fault zone structures along the rupture.”

“. Some available finite-fault slip models^{36-38,42} show relatively smooth slip variations across much of their fault models. Despite the differences, all models are characterized by a peak slip near the epicenter while showing a minor slip along the northeastern fault segment.”

I have moderate-to-major comments, mostly editorial and for clarification, which I list below (sort of sequential).

The finite-fault inversion approach is not well documented/explained, uncertainties are not stated, the weighting of the different data sets is not explained, the fault discretization etc. This makes it impossible to assess the robustness of the inferred models.

We added more text (see below) to clarify the finite-fault inversion method, uncertainties, the weighting of the different data sets, and fault discretization in the revision.

Results:

“Simulated annealing inversions frequently exhibit slight dependence on the chosen random seeds, mainly when multiple optimal solutions exist within the model space, exhibiting indistinguishable objective function values⁴⁴. Moreover, the varying random

seeds result in distinct initial fault models and Markov chains. To address this uncertainty and explore its impact, we conducted ten inversions for each event in the earthquake doublet using different random seeds in each case. The tests indicate that large-slip distributions of the ten models for the Mw 7.8 event exhibit relatively stable behavior, with consistency among the models (Supplementary Fig. 13a). In general, the standard deviation (STD) across most fault segments is negligible, with the exception of segments aF3 and aF6 (Supplementary Fig. 13b). Similarly, the STD for the Mw 7.7 event is typically small compared with the average slip (Supplementary Fig. 13c), but exceptions are found in the western bF1 and bF2 fault segments (Supplementary Fig. 13d). It is suspected that the higher STD in some parts of the fault model is caused by the absence of corresponding very near-fault observations, suggesting the need for further investigation in these areas."

Method:

"The sum of L1 and L2 norms of the seismograms in different wavelets quantifies the misfit between the recorded and synthetic waveforms. Sum-squared residuals have been adopted as the evaluation criteria to measure the difference between observed and synthetic static displacements. All inversions commence with a randomly generated initial model with a total moment equal to the GCMT solution. The weight assigned to the static error is set to be equal to the waveform error, but for the statics, the weight on the coseismic displacements derived from the strong-motion data is taken as half of GNSS statics accounting for the inherent uncertainties associated with baseline correction."

Abstract

+ L 23 — perhaps better ">59,000 fatalities;

We changed the text as suggested.

+ the abstract should tell which data are used and what has been done, before giving key results. This should be stated at the current end of Line 24, before rupture nucleation and propagation is presented. In some sense, Line 31-34 need to be moved forward, and then need to be expanded a bit

We revised the abstract as suggested.

Introduction

+ L 82: See above comment to Line 23

+ L 92: earth —> Earth

We changed the text, as suggested.

+ L 101: the authors here refer to the 2nd event to be of Mw 7.7. However, this has not been measured/inferred yet. In Line 80, they mention the USGS NEIC W-phase estimate of Mww 7.55, so this value of Mw 7.7 comes at a surprise and is confusing.

Many other studies report M 7.6 for the 2nd event. Please clarify.

The M_w 7.7 for the second event is based on the revised GCMT solution. We added a description in the introduction.

Kinematic Slip Models

+ L 131: "patchy heterogeneity" — somewhat awkward expression ... perhaps "spatial heterogeneity"?

We corrected it, as suggested.

+ L 133: unclear why variations in 'minor normal and thrust faulting' would be related to 'lateral inhomogeneities of crustal rocks' (assuming you mean the velocity-density structure here). Perhaps rephrase ...

We rephrased the statement (see below) that predominantly strike-slip motion accompanied by minor normal or thrust faulting indicates lateral inhomogeneities of tectonic stress and friction in the crust and fine fault zone structure along the rupture.

"The slip distribution exhibits significant spatial heterogeneity, characterized by predominant strike-slip motion with minor occurrences of normal or thrust faulting (Fig. 3a). This pattern aligns closely with published models³⁶⁻³⁸, highlighting the presence of lateral variations in tectonic stress, frictional properties within the crust, and intricate fault zone structures along the rupture."

+ L 154: remove 'oversimplified'

We removed the word, as suggested.

Triggering and Seismic Hazard Evaluation

+ This section header is misleading, as no seismic hazard calculation/evaluation is done. At most, there could be a "clock advance/delay". Please rephrase

We changed the title of this section to "Coseismic Coulomb stress changes and earthquake triggering effects"

+ L 190: add some references to the statement of 'significant variability in dip angles', and perhaps quantify this variability (5 degrees; 25 degrees?)

Figure 6 in the main text indicates the dip angle estimates from different seismological measures (used as target fault geometries), ranging from 42° to 86°, with either NW or SE plunge. We expand the text to discuss this in the revision.

“Due to the significant variability in the estimated dip angle for the larger event, with faulting geometries dipping to the northwest or to the southeast at angles from 42° to 86° being reported by different seismological institutes (Fig. 6),”

+ Line 195-201: Here it should be stated at which depth the CFS-values are calculated / reported. Is a stress increase of ~0.014 "significant"? And why is "~0.01 MPa" the minimal triggering threshold? This should be explained, and backup with references

The 10 km depth for the calculation is added to the text and indicated in Fig. 6. Previous studies have confirmed that a change in Coulomb stress value between 0.01 and 0.1 MPa can be enough to trigger a subsequent earthquake, with 0.01 MPa being a threshold value for earthquake-triggering. We add a reference to Ross Stein's work in the revision.

+ Line 202: seismic "risk" is not assessed in this study. Not even seismic hazard. See comment above.

Yes, we agree that we do not formally address risk or hazard per se, but we do quantify stress perturbations that are likely to influence future activity. This is clarified in the revision. It has been demonstrated by many studies that significant Coulomb stress changes can promote or inhibit subsequent earthquake occurrence. This is significant for earthquake interactions and seismic hazard evaluations, and can be utilized for assessing aftershock migration in the future. In our study, we identified notable increases in Coulomb stress along the Dead Sea Fault and along the northeastern segment of the EAF, which is important for future seismic risk/hazard assessments to consider.

+ Line 212: please quantify "incremental loading increase"

We added a specific value in the revision, as suggested.

“In A2, the receiver geometry of the left-lateral strike-slip Dead Sea fault, located just south of the mainshock rupture along the Amanos Fault, is calculated to have a loading increase (up to 0.1 MPa), suggesting an advance toward the next rupture.”

Discussion

+ Line 253: I find the argument on 'significant non-uniformity of stress accumulation' somewhat confusing, in particular when it is then linked to "high stress buildup" two lines below. Non-uniformity due to small-scale fault segmentation or fault roughness? Any evidence of this due to, say, variations in moment-tensor solutions in the local background seismicity there?

No major earthquakes have occurred on the EAF since the 19th century, and the background seismicity has produced relatively few recent focal mechanisms directly along the relevant section of the EAF. Historical studies referenced in the

manuscript (Duman & Emre, 2013, <https://doi.org/10.1144/SP372.14>; Güvercin et al., 2022, <https://doi.org/10.1093/gji/ggac045>) indicate that the EAF exhibits complex behavior, which is likely influenced by various factors, including oblique plate motion, variations in seismic coupling and fault maturity, and geometric complexities. Similar to determinations in other studies, our slip models clearly demonstrate a significantly non-uniform distribution in slip and rupture velocity, highlighting the uneven release of stress along the southwest section of the EAF. As we discuss, the EAF exhibits variations in stress orientations and the strain rate field, which is consistent with the diverse solutions for focal mechanisms. Our slip models show a non-uniformity of stress release along the fault, so we modify our language to "non-uniformity of stress release" instead of "significant non-uniformity of stress accumulation" in the revision.

+ Lines 260-280: this is interesting argument, which could be further quantified by computing the seismic-radiation efficiency. It is actually quite sobering and counterintuitive if these very damaging earthquakes had low radiation efficiency

Yes, the modest moment-scaled radiated energy is an important attribute of these events. Quantifying radiation efficiency also requires very reliable estimates of stress drop along with apparent stress, and for such a complex rupture, our confidence in stress drop values is low, as there is much data- and model-dependence of the estimates. At this time, we prefer not to pursue radiation efficiency estimation.

+ Line 285 - 298: This seems rather speculative, in my opinion, suggesting that super-shear rupture may even be common (or the norm?) for large earthquakes on the NAF and EAF. I don't think the case of prevalence of supershear event can be made, or even should be made, based on only four documented observations.

We modify the wording to make it less speculative, with only about a dozen convincing cases of supershear faulting, having 4 of them in Anatolia is quite a concentration, but we agree it is not a lot of cases.

Finite Fault Inversion

- + there is no mention of how the different datasets are weighted;
- + there is no mention of uncertainties at all;
- + there is no mention of how misfits are computed (variance reduction?)

=> this section needs to be expanded. The readers need information on robustness and uncertainties of the solutions

As suggested, we expanded this section in the revision (see below), to clarify the weights, uncertainties (revised Supplementary Fig 12), and misfits calculations in our finite fault inversions.

Results:

“Simulated annealing inversions frequently exhibit slight dependence on the chosen random seeds, mainly when multiple optimal solutions exist within the model space, exhibiting indistinguishable objective function values⁴⁴. Moreover, the varying random seeds result in distinct initial fault models and Markov chains. To address this uncertainty and explore its impact, we conducted ten inversions for each event in the earthquake doublet using different random seeds in each case. The tests indicate that large-slip distributions of the ten models for the MW 7.8 event exhibit relatively stable behavior, with consistency among the models (Supplementary Fig. 13a). In general, the standard deviation (STD) across most fault segments is negligible, with the exception of segments aF3 and aF6 (Supplementary Fig. 13b). Similarly, the STD for the MW 7.7 event is typically small compared with the average slip (Supplementary Fig. 13c), but exceptions are found in the western bF1 and bF2 fault segments (Supplementary Fig. 13d). It is suspected that the higher STD in some parts of the fault model is caused by the absence of corresponding very near-fault observations, suggesting the need for further investigation in these areas.”

Method:

“The sum of L1 and L2 norms of the seismograms in different wavelets quantifies the misfit between the recorded and synthetic waveforms. Sum-squared residuals have been adopted as the evaluation criteria to measure the difference between observed and synthetic static displacements. All inversions commence with a randomly generated initial model with a total moment equal to the GCMT solution. The weight assigned to the static error is set to be equal to the waveform error, but for the statics, the weight on the coseismic displacements derived from the strong-motion data is taken as half of GNSS statics accounting for the inherent uncertainties associated with baseline correction.”

Coulomb Stress

+ Line 418: "we calculated the triggering" ... this is actually not correct. Only the static stress changes were computed. It is then inferred/assumed that somewhat higher stress near the hypocenter may have initiated / facilitated the 2nd event. But there is dynamic rupture simulation carried out that actually shows that ...

Yes, we only calculated the static stress changes caused by the Mw 7.8 event, and found that the Mw 7.8 event stress change promoted the occurrence of the Mw 7.7 event. We rephrased the text.

References

+ several recent publications that also conducted finite-fault inversions are not listed, but should be added (see above)

We added several recent publications that have appeared since our paper was submitted in the revised manuscript.

Figures

+ Figure 3 and 4, panels: from which fault-normal distance were aftershocks projected onto the fault plane? Up to 5 km on either side of the fault plane? More or less?

Aftershocks less than 20 km on either side of the fault plane are projected onto the fault plane, but only the closest aftershocks are projected onto the fault plane for the intersecting faults. This is noted in the revised Fig. 3 caption.

+ Figures 8 and 9 are confusing: they show time-slices (snapshots) of the rupture process, but in each time slices, the rupture-time contours of the entire rupture is shown. Shouldn't the time-slice 0-10 sec only contain contours for the first 10 second?

Also, I find the color-range choice suboptimal, as tiny variations in shades of blue cannot be distinguished. Regions of zero slip should be shown in white, or light gray, and thus be clearly distinct from regions with 0.5-1.0 m. This is not case here.

We modified the figures as suggested in the revision.

Supplementary Material

Figure 3 is confusing here and should be removed. Maps related to the 2011 Tohoku earthquake are not needed here

We removed the original Supplemental Fig. 3 and added a link to a related database in the section of Code Availability.

We thank the reviewer for their very constructive comments, and the manuscript is improved by our revisions to address them. This is noted in the revised acknowledgments.

Reviewer #3 (Remarks to the Author):

Authors presented a thorough analysis on the rupture processes and possible impacts of the devastating Turkey earthquake doublet. The results suggest that both events ruptured multiple fault segments, featured supershear rupture episodes, and the first earthquake possibly aided the triggering of the second event. I like the idea of enriching the static displacement measurements using the near-field strong motion accelerograms. However, I have some concerns with the validity of the data product and results, as well as some unclear delivery of the method approach. Acknowledging that the implication of this work is important, here I suggest a major revision, see the following points for details.

Major comments:

1. The fault geometry is one major assumption and is critical to the finite fault

inversions conducted in this study. How the fault geometries, particularly the dip angles, are constructed, is unclear, which raises concerns about the validity of the findings. Additionally, the absence of satellite data and sources further compounds the issue.

We elaborate on the choices made regarding fault dip, which does vary amongst published models. Preliminary inversions with various strike/dip combinations for sparse fault models indicate typical limited direct waveform-based constraints on the dip for steeply-dipping strike-slip geometries. Thus, it is desirable to constrain the faulting geometry with a priori information to the extent possible. As discussed, we use the InSAR mapping to prescribe the active subfault locations and outcrop strikes along both ruptures. The distribution of seismicity with respect to the surface ruptures indicates that the dip direction of the EAF changes along strike. A detailed study of the aftershock activity of the Sivrice earthquake, just northeast of the Mw 7.8 rupture, indicates an NW steeply-dipping fault (Güvercin et al., 2022). This also holds for the northeastern part of the Mw 7.8 rupture based on cross-sections (D-D', and possibly C-C') of relocated aftershock as shown in new Supplementary Fig. 4. The aftershocks of the Mw 7.8 earthquake in the southwestern region (cross-sections A-A' and B-B') are mainly located southeast of the surface fault trace, indicating a SE dip, in agreement with the aftershock focal mechanisms in this zone. The along-strike change in fault dip may have influenced the bilateral rupture expansion. These relocated aftershock locations guide us in our choice of dip direction and angle, but there is clearly some uncertainty.

2. One of the major claims is the supershear stages found during both the northeast and southwest rupture episodes of the first event. However, how to validate (quantify the uncertainty) of such estimates? To date, most finite-fault inversions and back projection-based studies suggest subshear ruptures along the EAF during the first event. These includes:

Meng et al. (<https://doi.org/10.21203/rs.3.rs-2747911/v1>)

Mai et al. (<https://doi.org/10.1785/0320230007>)

Goldberg et al. (<https://doi.org/10.1785/0320230009>)

Melgar et al. (<https://doi.org/10.26443/seismica.v2i3.387>)

Among them, Meng et al. conducted a Mach Cone analysis arguing that the rupture is predominantly subshear.

We utilize a substantial amount of near-field strong-motion data, surpassing the data coverage of previous studies, to constrain the slip evolution of the Mw 7.8 event. Extensive testing was conducted on the rupture speed, revealing a notable contrast in the overall velocity of rupture propagation between the northeast (~ 3.5 km/s) and southwest (~2.5 km/s, but with the average rupture velocity on the aF5 segment being ~3.3 km/s, as depicted in Fig. 3a). This difference in rupture velocity was inferred by waveform fitting analysis, as documented in the Supplement. The relatively high rupture velocity observed in both directions is consistent with the preprint describing Meng's analysis and the teleseismic analysis of Mai et al. (2023). However, the BP analysis only provides an average rupture velocity for the

northeast and southwest directions, and does not resolve local variations in rupture velocity. This is especially true if there are delays in rupture between some segments, as appears to be the case for the Mw7.8 rupture, or if only a portion of the rupture is supershear (Mach wave coherence for surface waves will only be pronounced if there are long intervals of supershear rupture, which we do not think is the case). The excellent strong motion data coverage right along the fault allows us to constrain local intervals with variable rupture velocity with reasonable confidence.

3. Authors' slip models have significant slip gaps along the northeast segment of the East Anatolian Fault Zone, and the west of the Sürgü Fault. However, this appears to contradict the satellite surface displacement data which are continuous for both events (see Mai et al. Fig. 2; Goldberge et al. Fig. 4). How certain are these slip gaps? The quickest way is to do a forward prediction of the satellite displacement field and compare with the data.

To address the uncertainty and investigate its impact, we used different random seeds to perform ten inversions for the earthquake doublet. The results indicate that the large-slip distributions among the ten models for the Mw 7.8 event display relatively consistent behavior, demonstrating agreement among the models. Generally, the standard deviation (STD) across most fault segments is negligible, except for the aF3 and aF6 segments (see revised Supplementary Fig. 12), where higher variations are observed. Similarly, for the Mw 7.7 event, the STD is typically small compared to the average slip, with some exceptions found in the western parts of the bF1 and bF2 fault segments (revised Supplementary Fig. 12). The higher STD in these cases corresponds to slip gaps. We suspect that the elevated STD in certain areas of the fault segment is likely due to the lack of near-fault observations. This finding suggests the need for future investigation in these specific regions. We note that the slip models from teleseismic and geodetic data separately shown in Mai et al. (2023) differ significantly, but both display segment variability qualitatively similar to our inversions for the Mw 7.8 event. Our model is also similar to the Goldberg et al. (2023) model, which has substantial non-uniformity along strike. The along-strike variation in fault-parallel displacement inferred from pixel-tracking indeed has a more continuous distribution, possibly influenced by conditioning effects, but there are still rapid spatial slip variations of ~3-4 m over 40 km long sections, so there is substantial variability. Ongoing detailed mapping of surface ruptures along the fault zone may shed light on the true spatial heterogeneity. Besides, we did not include the pixel-tracking offsets in the joint inversions due several factors, including low resolution near the fault, the indistinguishability of coseismic displacements from two earthquakes, and potential early afterslip effects.

Overall, our analysis highlights the stability and consistency of the large slip distributions for this earthquake doublet, as inferred from the abundant seismic observations and static offset measurements. We also point out the presence of

localized variations and the need for additional research in certain areas where near-fault observations are lacking.

4. The conversion from strong motion accelerograms to static displacements impressively enriched the near-fault static observations. However, I am concerned with the robustness. For example, the static slip of station 4615 during the first earthquake has a 1.5 m amplitude and is perpendicular to the dominant strike slip direction of the East Anatolian Fault Zone, whereas the satellite radar images do not show any significant southeast motions (Mat et al, Goldberg et al.). Authors need to make sure that such conversions are valid.

We compared the localized coseismic displacements derived from strong-motion data and the horizontal displacements derived from pixel-tracking offsets of Sentinel-1 satellite radar images (Mai et al., 2023), finding overall reasonable agreement. However, there are exceptions observed at near-fault stations, as illustrated in new Supplementary Fig. 3. Given the inherent uncertainties associated with both approaches, such as the resolution of the pixels and the orientation error of the strong motion stations, these uncertainties inevitably contribute to differences in both magnitude and direction of the derived horizontal displacements. As for station 4614 and 4615 specifically, they exhibit high-quality strong-motion recordings without significant baseline shift other than correctable drift (see the following Figure), and has a strong fault-normal component for station 4615 (relative to the initial splay fault orientation) which is very similar to that of the pixel-tracking estimate (see revised Supplementary Fig. 3). A larger discrepancy between the estimates is seen at station 4614, but that site is likely influenced by contributions from slip on both the splay and the main EAF, so it is not unreasonable to have oblique motion. Therefore, using our new processing method, we have confidence in the stability and reliability of the static displacements derived from the strong-motion data.

Figure. Displacements integrated from strong-motion data at station 4614 and 4615 (black) with baseline corrections (red). The green curves indicate the estimated baseline effect.

“The strong-motion derived coseismic displacements are generally consistent with horizontal displacements derived from pixel-tracking offsets of Sentinel-1 satellite radar images³⁷. However, there are some differences at near-fault stations, as illustrated in Supplementary Fig. 3. Given the inherent uncertainties associated with both approaches, such as the resolution of the pixels and the orientation error of the strong motion stations, these uncertainties inevitably contribute to differences in both magnitude and direction of the derived horizontal displacements.”

5. 0.1 MPa is still small for static triggering of the Mw 7.7 second earthquake. Is there a source of minimal earthquake triggering threshold of ~ 0.01 MPa for M7 earthquakes?

Previous studies have confirmed that the change in Coulomb stress value between 0.01 and 0.1 MPa is thought to be enough to trigger an earthquake in the future, with 0.01 MPa being the threshold value for earthquake triggering. We added a reference to Ross Stein’s work in the revision. Triggerability is, of course, influenced by proximity to failure stress, so that very small peak dynamic strains can trigger failures (e.g., van der Elst and Brodsky, JGR, 2010).

Minor comments:

1. The colorbars for slip amount are severely saturated in Figures 3, 5, 8, 9. I can’t distinguish anything between 6 and 12 m.

We changed the colorbar to clarify the slip magnitude.

2. Still for these figures (especially Fig. 3 and 5), the slip models (subpanel a) are inversely oriented compared with the map (subpanel b). I had a hard time to realize that the left-hand side indicates the faults to the east.

We revised Figs. 3 and 5 to clarify the fault orientation.

We thank the reviewer for their comments, and the manuscript is improved by our revisions to address them. This is noted in the revised acknowledgments.

REVIEWERS' COMMENTS

Reviewer #3 (Remarks to the Author):

I would like to acknowledge that the authors have made a significant effort to address most of my concerns. The added comparison between aftershock distribution and fault dip justifies the selection of fault geometry. The additional test on multiple random seeds reflects the variation of slip distribution in the current settings and assumptions. Comparison between strong motion-derived displacements and the pixel tracking offsets illustrates the overall robustness of the derivation. Authors also improved their figures. Therefore, I recommend publication of the manuscript after evaluation of my following minor comments.

1. The overall high rupture velocity of the first event is consistent with other studies. The part I feel slightly concerned about is, the methodology from your slip inversion is essentially the same as Meng et al. and Goldberg et al., but the final kinematics has some (nontrivial) difference. Is this difference a reflection of the model uncertainty as well? Is it due to the imposed difference of assumptions on fault geometries and delays? Author may add one sentence in the discussion as a comment.

2. Although 0.1 MPa can induce seismicity, triggering of large destructive earthquakes are still much more rare than the occurrence of stress perturbation of 0.1 MPa. Is it somewhat related to the stochasticity of large earthquake nucleation? Author may add one sentence in the discussion as a comment.

Response to reviews of "Complex multi-fault rupture and triggering during the February 6, 2023, earthquake doublet in southeastern Türkiye" by Liu et al., submitted to Nature Communications. The review comments are reproduced below in black type, with our responses and indications of how we have revised the manuscript to address the reviews indicated in blue.

REVIEWERS' COMMENTS

Reviewer #3 (Remarks to the Author):

I would like to acknowledge that the authors have made a significant effort to address most of my concerns. The added comparison between aftershock distribution and fault dip justifies the selection of fault geometry. The additional test on multiple random seeds reflects the variation of slip distribution in the current settings and assumptions. Comparison between strong motion-derived displacements and the pixel tracking offsets illustrates the overall robustness of the derivation. Authors also improved their figures. Therefore, I recommend publication of the manuscript after evaluation of my following minor comments.

1. The overall high rupture velocity of the first event is consistent with other studies. The part I feel slightly concerned about is, the methodology from your slip inversion is essentially the same as Meng et al. and Goldberg et al., but the final kinematics has some (nontrivial) difference. Is this difference a reflection of the model uncertainty as well? Is it due to the imposed difference of assumptions on fault geometries and delays? Author may add one sentence in the discussion as a comment.

Yes, there is variation among the published kinematic models, including one presented in the new paper by Jia et al. that we now cite. It is challenging to specifically state exactly why there are differences, but in general, these will occur due to differences in precise model geometries (different dips are used for some fault segments between studies), different data (for example, our study is the only to be using the expanded near-fault static displacement observations obtained from strong-motion instruments to constrain the coseismic slip), along with differences in inversion algorithms and model parameterization. We added a sentence to note these issues, as suggested.

2. Although 0.1 MPa can induce seismicity, triggering of large destructive earthquakes are still much more rare than the occurrence of stress perturbation of 0.1 MPa. Is it somewhat related to the stochasticity of large earthquake nucleation? Author may add one sentence in the discussion as a comment.

Indeed, triggering intuitively requires pre-stress accumulation that has approached failure, and only then can ~ 0.01 MPa stress increments that are favorably oriented initiate rupture onset. The empirically estimated lower stress increment level of ~ 0.01 MPa increment may reflect observational limits, but is a threshold consistent with many observations. We add a brief discussion of this issue as suggested.